# Boosting In-Silicon Directed Evolution with Fine-Tuned Protein Language Model and Tree Search

## Abstract

Protein evolution through amino acid sequence mutations is a cornerstone of life sciences. While current in-silicon directed evolution algorithms largely focus on designing heuristic search strategies, they overlook how to integrate the transformative protein language models, which encode rich evolutionary patterns, with reinforcement learning to learn to directly evolve proteins. To bridge this gap, we propose AlphaDE, a novel framework to optimize protein sequences by harnessing the innovative paradigms of large language models such as fine-tuning and test-time inference. First, AlphaDE fine-tunes pretrained protein language models using masked language modeling on homologous protein sequences to activate the evolutionary plausibility for the interested protein class. Second, AlphaDE introduces test-time inference based on Monte Carlo tree search, which effectively evolves proteins with evolutionary guidance from the fine-tuned protein language model. Extensive benchmark experiments show that AlphaDE remarkably outperforms previous state-of-the-art methods even with few-shot fine-tuning. A further case study demonstrates that AlphaDE supports condensing the protein sequence space of avGFP through computational evolution.

## 1 Introduction

Proteins are essential components of living systems, exhibiting a great diversity of functions among biological macromolecules (Holm & Sander, 1996). They play critical roles in a wide range of biochemical processes, including enzyme catalysis, cellular metabolism, immune responses, and signal transduction (Jiang et al., 2024a). Protein engineering through directed evolution enables optimization of protein functions by seeking potential protein variants with improved properties such as the expression level and catalytic activity (Yang et al., 2019). Traditional in vitro or in vivo experimental directed evolution approaches, such as deep mutational scanning (Fowler & Fields, 2014) and orthogonal DNA replication system (Ravikumar et al., 2014), directly measure the functional effects of protein mutations but are limited to exploring only a fraction of the possible protein space, and are usually expensive and laborious. To reduce the burden of the expensive wet experiments, recent advances in machine learning-guided approaches focus on building a surrogate sequence-function landscape (Yang et al., 2019), and a lot of in-silicon directed evolution algorithms (Brookes & Listgarten, 2018; Sinai et al., 2020; Ren et al., 2022; Wang et al., 2023) have been proposed to strategically explore the protein fitness landscape to identify optimal sequence mutations via an iterative search process.

On the other hand, protein language models (Meier et al., 2021; Lin et al., 2023), which encapsulate millions of years of evolutionary information through unsupervised pretraining in massive protein databases, are rapidly changing the domain of protein design (Madani et al., 2023; Jiang et al., 2024b; Hayes et al.,

2025). As protein language models implicitly learn complex evolutionary and structural dependencies from natural protein sequences, researchers have been increasingly employing protein language models for protein engineering tasks such as zero-shot inference of the functional effects of sequence substitutions to find high-fitness variations (Hie et al., 2024; Shanker et al., 2024). More recently, some pioneering works have begun to introduce pretrained protein language models in the process of directed evolution (Jiang et al., 2024b; Yang et al., 2025b; Tran & Hy, 2025). However, although employing protein language models to recommend mutations, these works use simple heuristic search methods such as greedy selection (Jiang et al., 2024b) or beam search (Tran & Hy, 2025) to search in directed evolution, leaving a large room to explore how to integrate protein language models into advanced optimization techniques such as reinforcement learning.

To fill this gap, in this work, we propose a novel framework named AlphaDE to directly evolve protein sequences following the technical paradigm of natural large language models (Guo et al., 2025). Specifically, AlphaDE consists of a fine-tuning step and a Monte Carlo tree search (MCTS) inference step built on the protein language model. In the fine-tuning step, AlphaDE fine-tunes the pretrained protein language model with homologous protein sequences to activate its evolutionary plausibility for the interested protein class. In the MCTS-assisted inference step, similar to test-time MCTS in large language models to boost reasoning (Zhou et al., 2024; Guan et al., 2025), AlphaDE conducts an iterative tree search to efficiently optimize protein function via residue mutations guided by the fine-tuned protein language model. Through the two synergic steps, AlphaDE harvests superior directed evolution ability for proteins. To evaluate AlphaDE, we conduct computational benchmark experiments on eight distinct tasks. Impressively, AlphaDE substantially outperforms various in-silicon directed evolution methods. Further few-shot fine-tuning experiments reveal that AlphaDE's evolution ability can be activated by fine-tuning with dozens of homologous sequences. Lastly, to show the broader applicability of AlphaDE, we conduct a proof-of-concept task to computationally condense the sequence space of the functional avGFP protein.

## 2 BACKGROUND

### 2.1 PROTEIN LANGUAGE MODELS

In analogy to large language models in natural language processing, protein language models such as ESM (Rives et al., 2021; Lin et al., 2023), ProteinBERT (Brandes et al., 2022), and ProGen (Madani et al., 2023) have surged in modeling protein sequences. Pretrained by masked language modeling (Devlin et al., 2019) or autoregressive language modeling (Brown et al., 2020) on evolutionary-scale protein databases, protein language models have demonstrated outstanding potential in protein design. For example, recently, Hayes et al. (2025) prompt a protein language model (i.e., ESM3) to generate a bright fluorescent protein far different from known fluorescent proteins, which is estimated to simulate five hundred million years of evolution. Another notable example is that Bhat et al. (2025) utilize contrastive language modeling to design peptide binders to conformationally diverse targets using only the amino acid sequence of the target protein.

At the same time, some works (Widatalla et al., 2024; Yang et al., 2025a) utilize labeled sequence-fitness pairs to steer protein language models for *de novo* protein sequence design to generate novel and high-quality sequences, showing the superior alignment ability of protein language models.

### 2.2 PROTEIN DIRECTED EVOLUTION

Directed evolution is a classical paradigm for protein sequence design, where a plenty of algorithms are developed to accelerate the in-silicon directed evolution process. AdaLead (Sinai et al., 2020) is an advanced implementation of model-guided directed evolution with iteratively recombined and mutated operations for seed sequences. CMA-ES (Hansen & Ostermeier, 2001) is a second-order evolutionary search algorithm that estimates the covariance matrix to adaptively adjust the search strategy of the upcoming generations.

Bayesian optimization (BO) (Snoek et al., 2012) is a classical paradigm for the sequential design problem (Mockus, 1989), which estimates the uncertainty and constructs the acquisition function for exploration. DbAS (Brookes & Listgarten, 2018) establishes a probabilistic framework that trains a variational autoencoder (VAE) (Kingma & Welling, 2022) to model the distribution of high-fitness sequences and adaptively samples sequences from this trained VAE to explore the fitness landscape. Following DbAS, CbAS (Brookes et al., 2019) estimates the probability distribution conditioned on the desired properties with model-based adaptive sampling and additionally considers a regularization to stabilize the model-guided search process. DyNA-PPO (Angermueller et al., 2020) formulates protein sequence design as a sequential decision-making problem and uses proximal policy optimization (PPO) (Schulman et al., 2017) to perform sequence generation. PEX (Ren et al., 2022) aims to search for effective candidates of low-order mutants near the wild-type, and formulates this process as a proximal optimization problem to solve. EvoPlay (Wang et al., 2023) uses the self-play reinforcement learning inspired by AlphaZero (Silver et al., 2018) to optimize protein sequences. Recently, TreeNeuralTS and TreeNeuralUCB (Qiu et al., 2024) combine tree search with bandit machine learning for directed evolution, which expands a tree starting from the initial sequence with the guidance of a bandit machine learning model. Specifically, TreeNeuralTS adopts Thompson sampling (Thompson, 1933) while TreeNeuralUCB utilizes the upper confidence bound (Auer et al., 2002) to explore the sequence space.

More recently, with the emergence and development of protein language models, some pioneering works explore to integrate the pretrained protein language models in the process of directed evolution (Jiang et al., 2024b; Yang et al., 2025b; Tran & Hy, 2025). For example, Jiang et al. (2024b) and Yang et al. (2025b) present the multi-round active learning to rapidly improve protein activity, which alternate between collecting sequence fitness using a wet-lab assay and training a protein language model to prioritize new sequences to screen in the next round. Meanwhile, LatentDE (Tran et al., 2025) uses gradient-based method to search in the latent space represented by VAE with protein embeddings extracted from a frozen, general-purpose pretrained protein language model (i.e., ESM-2). Moreover, MLDE (Tran & Hy, 2025) introduces an optimization pipeline that utilizes protein language models to pinpoint the mutation hotspots and then suggest replacements by heuristically calculating a k-mer's relevancy and entropy in a sequence. However, despite leveraging protein language models to recommend mutations, these works use simple search methods such as greedy selection (Jiang et al., 2024b) or beam search (Tran & Hy, 2025), remaining how to integrate protein language models into advanced optimization techniques such as reinforcement learning unexplored in the directed evolution field. To bridge this gap, here we follow a principle of effective searching (i.e., AlphaZero-like MCTS) with strong prior guidance (i.e., fine-tuned protein language model), inspired by the natural large language model technique paradigm such as fine-tuning (Shao et al., 2024) and test-time inference (Guan et al., 2025), to boost in-silicon directed evolution with a novel technique paradigm.

## 3   ALPHADE

In this section, we show AlphaDE for protein directed evolution in detail. First, we give the problem definition of the directed evolution in Section 3.1. Second, in Section 3.2, we introduce how to fine-tune the pretrained protein language model to give prior guidance on the next mutation residues. Third, in Section 3.3, we show how to perform tree search to directly evolve protein sequences following the prior mutation guidance from the fine-tuned protein language model. The whole framework of AlphaDE is shown in Figure 1.

### 3.1   PROBLEM DEFINITION

Protein directed evolution can be formulated as a Markov decision process (MDP) (Bellman, 1957), given that the next mutation residue to be chosen depends only on the current protein sequence (Wang et al., 2023). The MDP is defined as $M = (S, A, P, R)$ where $S$ denotes the set of states that describe the current protein sequence, $A$ denotes the set of actions that indicate the chosen position and residue type to be mutated from

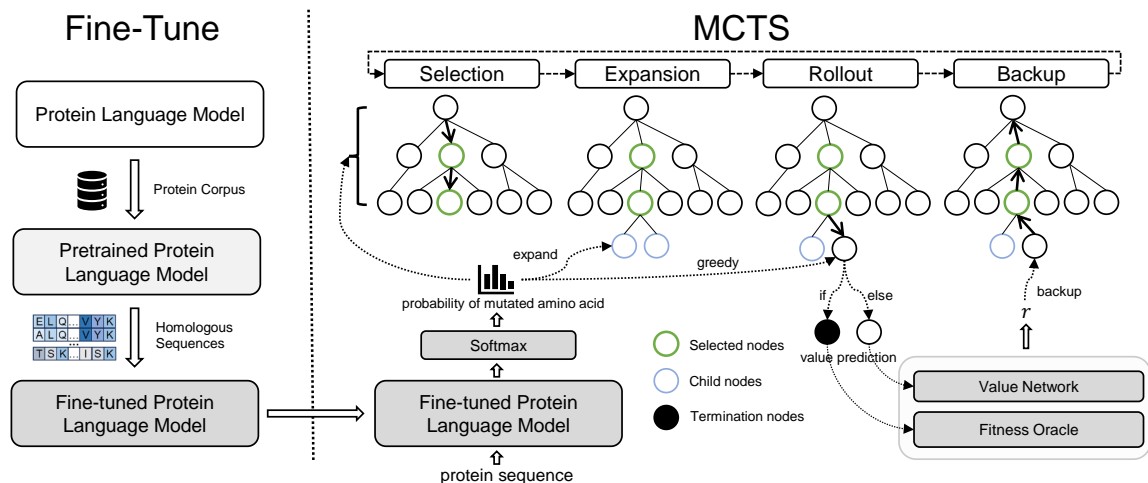

Figure 1: The framework of AlphaDE. It consists of a fine-tuning step and an MCTS inference step.

the current protein sequence, and $P : S \times A \to S$ is the state transition function where the current protein sequence incorporates the chosen residue position and type to mutate to a new protein sequence. The protein sequence state $s \in S$ is represented by a binary matrix of size $20 \times L$ with columns indicating positions ($L$) and rows indicating amino acids (20). Correspondingly, the action $a \in A$ is a flattened one-hot vector of size $20 \times L$. When an action $a$ is performed, the element of value 1 in the action vector is selected, resulting in a single-site mutation on the current protein sequence $s$. The element indicated by the mutation action in the state matrix is changed to 1, while the other elements (residue types) of the same column (position in protein) are all set to 0. An episode contains a series of mutation actions $a_t$ and states $s_t$ where $0 < t < T$ and $T$ is the termination step. $R : S \to \mathbb{R}$ is the episodic reward function indicating protein fitness, such as binding affinity, which can be accessed by an experimental assay or predicted by a simulated oracle. Directed evolution aims to take actions that maximizes $\overline{r}$, which is approximated under repeated rollouts (Gelly & Silver, 2011) as

$$\overline{r}(s,a) = \frac{1}{N(s,a)} \sum_{j=1}^{N(s)} \mathbb{I}_j(s,a) r_j(s),$$ (1)

where $N(s)$ denotes the rollout times starting from state $s$ and $N(s,a)$ is the times that action $a$ has been taken from state $s$. $\mathbb{I}_j(s,a)$ is an indicator function with value 1 if action $a$ is selected from state $s$ at the $j$th rollout round, 0 otherwise. $r_j(s)$ is the episodic reward of the final protein sequence at the terminal state for the $j$th rollout round starting from state $s$. A larger $\overline{r}(s,a)$ indicates a higher expected episodic reward by taking action $a$ from state $s$. The measurement of episodic reward $r_j(s)$ usually relies on wet experiments, which are time-consuming and expensive. Therefore, in-silicon directed evolution simulates the ground-truth protein fitness landscape by a proxy oracle model to replace the wet-lab measurements.

## 3.2 FINE-TUNING PRETRAINED PROTEIN LANGUAGE MODEL

Fine-tuning natural large language models has become a standard step to better solve specific tasks such as math (Shao et al., 2024), coding (Muennighoff et al., 2024), and medical analysis (Christophe et al., 2024) to strengthen the specialized ability pretrained in the large language models. Inspired by this, we fine-tune the protein language model with homologous protein sequences to activate its evolutionary plausibility for the interested protein class. Specifically, AlphaDE uses the unsupervised masked language modeling objective

(Devlin et al., 2019) without any fitness label. During fine-tuning, each input protein sequence is corrupted by replacing a fraction of amino acids with a special mask token. The protein language model is then trained to predict masked tokens in the corrupted sequences sampled from a homologous sequence set $S_h$:

$$L_{mlm} = \mathbb{E}_{s \sim S_h} \mathbb{E}_M \sum_{i \in M} -log p(s_i|s_{/M}). \tag{2}$$

For each sequence $s$, we sample a set of indices $M$ to mask, replacing the true token at each index $i$ with the mask token. For each masked token, we independently minimize the negative log likelihood of the true amino acid $s_i$, given the masked sequence $s_{/M}$ as context. Intuitively, to predict a masked residue, the protein language model must identify dependencies between the masked site and the unmasked parts of the sequence, therefore capturing the co-evolution information among residues. After the fine-tuning step, the protein language model recovers masked protein sequences towards the possible existing sequences of the same class as homologous sequences, which gives a prior distribution of next mutations for the MCTS step.

### 3.3 MONTE CARLO TREE SEARCH WITH FINE-TUNED PROTEIN LANGUAGE MODEL

We use Monte Carlo tree search (MCTS) to solve the MDP of directed evolution. To effectively select actions, some MCTS works (Rosin, 2011; Silver et al., 2016) train neural networks based on expert data for experienced guidance. Similarly, to enable efficient guidance of protein mutation, AlphaDE employs a fine-tuned protein language model to predict the next mutation action (the residue position and residue type) given the current protein sequence. At each step, the fine-tuned protein language model receives the protein sequence and outputs the probability logits of each residue type on each residue position as guidance for MCTS.

When mutating protein sequences with the fine-tuned protein language model, although we could mutate a protein sequence in a greedy manner by taking the next mutation with the maximum probability, it is prone to being stuck in a local optimum due to the unpredictable complexity of the protein fitness landscape (Romero & Arnold, 2009). Moreover, the predicted mutation with the maximum probability does not mean that it is optimal because the protein language model is not optimized for a specific protein fitness. To directly maximize a given protein fitness, we integrate MCTS into AlphaDE to solve the MDP of directed evolution with the help of a fine-tuned protein language model for mutation guidance. Next, we introduce the MCTS part of AlphaDE in the background of protein directed evolution.

MCTS (Browne et al., 2012) adopts a tree structure to perform simulation iterations and estimate the state value of actions. Meanwhile, it uses the previously estimated action values to guide the search process towards higher rewards. As shown in Figure 1, the MCTS part in AlphaDE consists of the following four steps per iteration:

- **Selection**. Each iteration starts from sequence $s_\tau$, recursively selecting the best child node until reaching a leaf node $a_{\tau+l}$, i.e., a node that has not been expanded or terminated, after $l$ selections. At each selection step $t \in [1, l]$, a selection criterion determines the best child node to be chosen, which balances exploitation and exploration to avoid being trapped in local optimums. We use Predictor with Upper Confidence bounds applied to Trees (PUCT) (Rosin, 2011) as the selection criterion for each candidate child node as

$$U_{puct}(s_{\tau+t-1}, a) = \frac{W_a}{N_a} + c P_{plm}(a|s_{\tau+t-1}) \frac{\sqrt{N}}{1 + N_a}, \tag{3}$$

where the constant $c$ controls the exploration degree. $W_a$ is the cumulative reward of node $a$, $N$ is the total visit count, and $N_a$ is the visit count for node $a$. The term $\frac{\sqrt{N}}{1+N_a}$ encourages selection of less-visited nodes, while $P_{plm}(a|s_{\tau+t-1})$ favors beneficial mutations predicted by the fine-tuned protein language model. Furthermore, $\frac{W_a}{N_a}$ promotes exploitation of the nodes with high protein function fitness.

- **Expansion**. Given a selected leaf node $a_{\tau+l}$, the fine-tuned protein language model computes the probability $P_{plm}(a|s_{\tau+l})$ for each expandable action $a \in A$ as a prior distribution over mutations. Here, $s_{\tau+l}$ is the state context of node $a_{\tau+l}$, and $A$ denotes the legal action space, i.e., the residue type and position in protein. The expanded child nodes of the leaf node $a_{\tau+l}$ are immediately added and initialized in the tree.

- **Rollout**. The value of the reached leaf node $a_{\tau+l}$ is evaluated by a rollout. From the leaf node, MCTS recursively mutates the sequence until termination, then the final sequence $s_T$ is evaluated by a fitness oracle for reward $r$. During the rollout, each mutation is selected greedily. To speed up the computationally expensive rollout process, AlphaDE trains a value network to predict the state value of the reached leaf nodes as AlphaZero (Silver et al., 2018). Specifically, if the reached leaf node $s_{\tau+l}$ is a termination node, it is evaluated by the fitness oracle. Otherwise, it is predicted by the value network.

- **Backup**. After rollout, the final reward $r$ is backpropagated along the visited nodes to update their statistics until the root node. The detailed updating for tree nodes is elaborated as

$$N_a \leftarrow N_a + 1, W_a \leftarrow W_a + r, a \leftarrow \text{parent of } a, \tag{4}$$

where $N_a$ is the visit count and $W_a$ is the cumulative reward of node $a$. For each selection $t \in [1, l]$, the statistics of node $a_{\tau+t}$ are updated by adding the rollout reward of $a_{\tau+l}$ to $W_a$ and increasing $N_a$ by 1.

## 4 EXPERIMENTS

In this section, we conduct computational experiments to validate the effectiveness of AlphaDE. First, in Section 4.1, we benchmark AlphaDE with various directed evolution algorithms on extensive tasks. Second, we study AlphaDE under different settings of homology sequence availability, such as few-shot fine-tuning in Section 4.2 and zero-shot scaling with model sizes in Section 4.3. Finally, we utilize AlphaDE to computationally condense the protein sequence space of the fluorescent protein in Section 4.4. The ablation study of AlphaDE is provided in Appendix F.

### 4.1 BENCHMARK EXPERIMENTS

We benchmark AlphaDE and baselines on a suite of eight in-silicon protein engineering tasks (Ren et al., 2022). These tasks involve extensive protein applications such as biosensor design and industrial enzyme renovation. More descriptions about the eight benchmark tasks are provided in Appendix A. Following previous works (Ren et al., 2022; Wang et al., 2023; Qiu et al., 2024), we simulate the ground-truth fitness landscape of protein with an oracle model TAPE (Rao et al., 2019) to replace wet laboratory measurements.

We compare AlphaDE with various baselines including LatentDE, MLDE, TreeNeuralTS, TreeNeuralUCB, PEX, EvoPlay, AdaLead, DyNA-PPO, DbAS, CbAS, BO, and CMA-ES. We follow the officially released codebase and configurations of LatentDE, MLDE, TreeNeuralTS, TreeNeuralUCB, and EvoPlay for running. We take the results of PEX (one version with CNN model and another with MuFacNet model), AdaLead, CbAS, DbAS, and DyNA-PPO from Ren et al. (Ren et al., 2022) as we use the same benchmark setting, such as the starting sequence and fitness oracle. We run BO and CMA-ES with the implementations in FLEXS (Sinai et al., 2020). We average the results of AlphaDE, LatentDE, MLDE, TreeNeuralTS, TreeNeuralUCB, EvoPlay, BO, and CMA-ES over 5 independent trials. For all methods, we set the same oracle budget of 1000 queries. For AlphaDE, here we use the pretrained ESM2-series models (Lin et al., 2023) for fine-tuning as it is one of the most powerful and widely used protein language models with various model sizes for study convenience. Specifically, in the standard AlphaDE, we use ESM2-35M for fine-tuning as its efficiency in model size and effectiveness in evolution plausibility. Details of the homologous sequence datasets for fine-tuning are in Appendix B, while hyperparameters of AlphaDE are given in Appendix C. For each task, the starting protein sequence for optimization is the sequence with the lowest protein fitness value, which is the same as the starting sequences in TAPE oracles. The benchmark results for these eight tasks are summarized in Table 1.

Table 1: Benchmarking in-silicon directed evolution methods. We present the maximum fitness in 1000 black-box oracle queries.

| Method | avGFP | AAV | TEM | E4B | AMIE | LGK | PAB1 | UBE2I |
|---|---|---|---|---|---|---|---|---|
| AlphaDE | **3.86** | **7.95** | **1.22** | **7.75** | **0.24** | **0.04** | **1.47** | 2.97 |
| LatentDE (Tran et al., 2025) | 3.79 | 4.72 | 1.20 | 3.88 | -3.34 | -1.13 | 0.58 | 1.29 |
| MLDE (Tran & Hy, 2025) | 2.04 | -3.38 | 0.08 | 3.84 | -1.38 | -0.19 | 0.92 | 2.75 |
| TreeNeuralTS (Qiu et al., 2024) | 2.44 | 2.47 | 0.27 | 0.79 | -0.22 | **0.04** | 1.02 | **2.98** |
| TreeNeuralUCB (Qiu et al., 2024) | 2.37 | 3.85 | 0.19 | 0.70 | -0.19 | **0.04** | 1.27 | 2.95 |
| PEX (MuFacNet) (Ren et al., 2022) | 3.12 | 4.45 | 0.27 | 2.22 | 0.16 | **0.04** | 1.23 | 2.97 |
| PEX (CNN) (Ren et al., 2022) | 2.97 | 2.52 | 0.19 | 2.21 | -0.11 | 0.03 | 1.27 | 2.97 |
| EvoPlay (Wang et al., 2023) | 1.72 | -3.45 | 0.01 | -0.40 | -0.88 | -1.09 | 0.34 | 1.87 |
| EvoPlay (Wang et al., 2023) | 1.72 | -3.45 | 0.01 | -0.40 | -0.88 | -1.09 | 0.34 | 1.87 |
| AdaLead (Sinai et al., 2020) | 2.61 | -2.33 | 0.09 | 0.16 | -0.86 | -0.72 | 1.09 | 2.91 |
| DyNA-PPO (Angermueller et al., 2020) | 1.84 | -3.22 | 0.03 | -0.20 | -2.13 | -0.32 | 0.42 | 2.17 |
| DbAS (Brookes & Listgarten, 2018) | 2.30 | -2.43 | 0.10 | 0.23 | -2.30 | -0.35 | 0.90 | 2.85 |
| CbAS (Brookes et al., 2019) | 2.22 | -2.50 | 0.10 | 0.19 | -2.26 | -0.22 | 0.92 | 2.87 |
| BO (Snoek et al., 2012) | 1.57 | -3.84 | 0.02 | -0.33 | -5.71 | -1.45 | 0.39 | 0.02 |
| CMA-ES (Hansen & Ostermeier, 2001) | 1.60 | -3.50 | 0.02 | -0.09 | -7.92 | -1.29 | 0.53 | -0.01 |

The results in Table 1 show that AlphaDE significantly outperforms various baselines in most tasks. For example, in the task of AAV, the second best method MLDE obtains a fitness value of 4.72 while AlphaDE achieves 7.95 with an improvement of 68.43%. Similar situations also happen in the task of E4B and AMIE, which exhibit AlphaDE's great evolution ability by combining effective searching (i.e., AlphaZero-like MCTS) with strong prior guidance (i.e., fine-tuned protein language model). At the same time, we also examine the diversity of evolved sequences by AlphaDE in Appendix I. AlphaDE with different ESM2 model sizes is tested in Appendix D. The ablation study of AlphaDE on the fine-tuning step and MCTS inference step is given in Appendix F. Benchmark results using another oracle based on ESM-1b further validate AlphaDE as shown in Appendix G. Moreover, the study of hyperparameter sensitivity of AlphaDE is available in Appendix H, while AlphaDE's computation efficacy is provided in Appendix L.

At the same time, we also test AlphaDE with other protein language models such as ProtBert (Elnaggar et al., 2022) (model size 420M) and ESM-1b (Rives et al., 2021) (model size 650M), which are trained with the standard BERT (Devlin et al., 2019) architecture using the UniRef100 (Suzek et al., 2007) database and with the optimized RoBERTa (Liu et al., 2019) architecture using the UniRef50 (Suzek et al., 2007) database, respectively. The results of AlphaDE with ProtBert and ESM-1b are shown in Table 2. We see that AlphaDE with ProtBert and AlphaDE with ESM-1b perform differently in some tasks, suggesting their evolution ability varies in different protein families. Generally, AlphaDE with the two different protein language models achieves superior performance, showing AlphaDE's compatibility. Meanwhile, the fine-tuning step is the key to boosting performance, as directly using the pretrained versions of protein language models leads to a lot of fitness loss.

Table 2: Results of AlphaDE with ProtBert and ESM-1b models. We present the maximum fitness scores obtained in 1000 black-box oracle queries. Results are averaged over five independent trials.

| AlphaDE Model | avGFP | AAV | TEM | E4B | AMIE | LGK | PAB1 | UBE2I |
|---|---|---|---|---|---|---|---|---|
| fine-tuned ProtBert (Elnaggar et al., 2022) | 3.09 | 7.66 | 0.49 | 7.66 | 0.00 | -0.01 | 0.41 | 2.71 |
| w/o fine-tuning | 1.53 | -0.94 | 0.24 | -0.37 | 0.02 | 0.00 | 0.19 | 1.36 |
| fine-tuned ESM-1b (Rives et al., 2021) | 3.09 | 16.97 | 0.49 | 7.85 | 0.03 | 0.01 | 0.60 | 1.49 |
| w/o fine-tuning | 1.46 | -0.96 | 0.24 | 2.10 | -0.53 | 0.00 | 0.42 | 1.47 |

### 4.2 FEW-SHOT FINE-TUNING EXPERIMENTS

One of the biggest advantages of large language models is that they are few-shot learners (Brown et al., 2020). Here we investigate whether this advantage is held in the protein language model for protein directed evolution tasks. We use the ESM2-35M model as the base model for few-shot fine-tuning and fine-tune it with different numbers of protein sequences including 16, 32, 64, 128, 256, 512, and 1024. These protein sequences for few-shot fine-tuning are randomly sampled from the whole dataset and are used to fine-tune ESM2-35M with 3 epochs. The results of AlphaDE with different few-shot fine-tuned ESM2-35M models are given in Figure 2. We see that, even with dozens of protein sequences, the evolution ability of AlphaDE could be greatly improved, and the performance increases with the number of fine-tuning protein sequences.

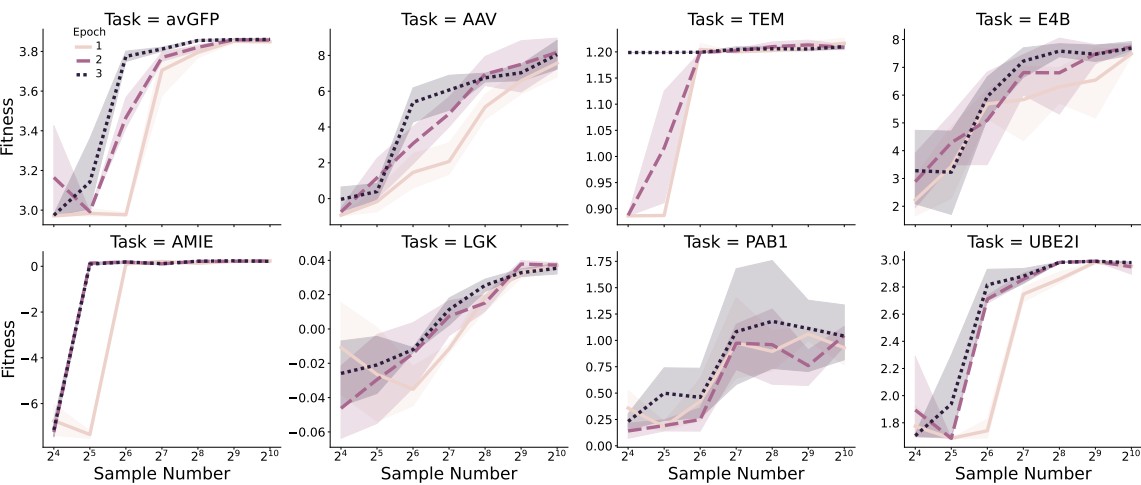

Figure 2: AlphaDE with fine-tuned ESM2-35M, which are fine-tuned with different numbers of sequences randomly sampled from the whole data distribution. 95% confidence intervals are shadowed.

Although AlphaDE does not utilize fitness labels of protein sequences in the pretrain, fine-tune, or MCTS inference, we additionally test whether the protein fitness influences the performance of AlphaDE by fine-tuning on the sampled sequences from the bottom 20% of data with the lowest fitness. The results of fine-tuning with low-fitness data are plotted in Figure 3 and are very similar to Figure 2, showing that AlphaDE's evolution capability is unaffected by fitness values during fine-tuning. This suggests applicability in real-world settings where only sequences are available, without costly fitness measurement assays. We also show that AlphaDE works with the sequences from homology searching such as HMMER (Potter et al., 2018) in Appendix J.

### 4.3 ZERO-SHOT SCALING EXPERIMENTS ON MODEL SIZE

Next, we study how AlphaDE scales with pretrained protein language model sizes in a zero-shot setting, testing ESM2 models from 8M to 15B. Results in Figure 4 show that, in most tasks except avGFP and AAV, AlphaDE's performance roughly increases with pretrained model sizes. This is different from Table 3, where the fine-tuned model size has little effect. Interestingly, with the largest pretrained 15B model, AlphaDE's performance matches with its fine-tuned ESM2-35M version on tasks of TEM, AMIE, and UBE2I. However, in most tasks, the pretrained models, even 15B, perform worse than the standard AlphaDE with fine-tuned ESM2-35M. We see two points here. First, larger pretrained models encode more evolutionary information,

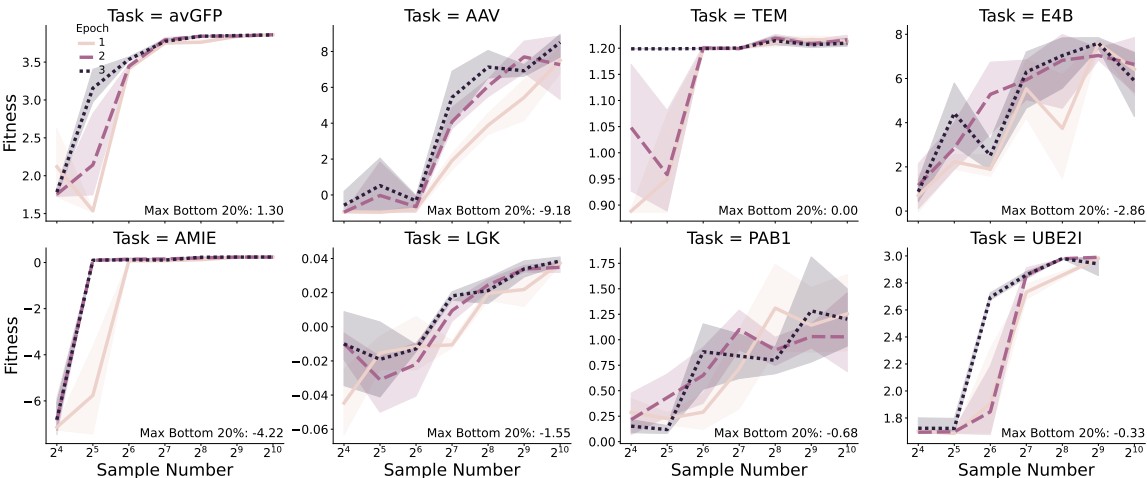

Figure 3: AlphaDE with fine-tuned ESM2-35M, which are fine-tuned with different numbers of sequences randomly sampled from the bottom 20% data. The "Max Bottom 20%" value denotes the maximum fitness value in the bottom 20% data. 95% confidence intervals are shadowed.

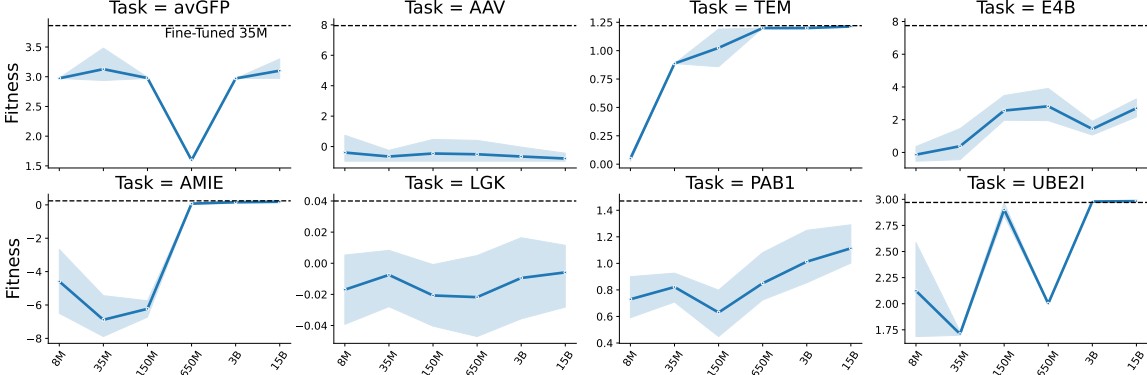

Figure 4: AlphaDE's performance scales with sizes of pretrained protein language models. The black dashed line indicates AlphaDE with fine-tuned ESM2-35M. 95% confidence intervals are shadowed.

enhancing AlphaDE's evolution ability in most cases. Second, fine-tuned small models with homologous sequences provide strong task-specific evolution ability, underscoring the necessity of fine-tuning.

## 4.4 CONDENSING PROTEIN SEQUENCE SPACE

The origin and evolution of protein folds remain a fundamental challenge in biology (Chothia & Gerstein, 1997; Levitt, 2009), with methods for exploring evolutionary trajectories still underdeveloped. Leveraging AlphaDE's evolutionary capability, we address this through condensing the avGFP sequence space by evolving an incomplete, non-folding sequence into a functional, folded protein. Inspired by hypotheses that proteins evolve from small random peptides into complex folded structures (Nepomnyachiy et al., 2017; Kolodny et al., 2021), we keep the chromophore and $\beta$-barrel residues of avGFP, mask half the remaining

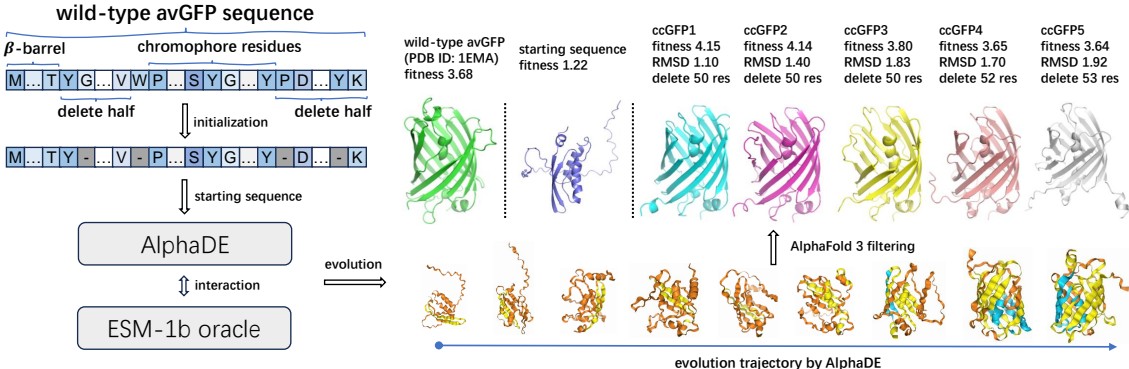

Figure 5: The illustrated process of AlphaDE to condense the sequence space of avGFP. The evolution trajectory is sampled during one trial, and the first is the predicted structure of the starting sequence.

sequence, and use AlphaDE to optimize the predicted fluorescence fitness to recover protein function. If AlphaDE recovers the predicted fluorescence function with fewer amino acid residues than the wild-type avGFP sequence, it successfully condenses the protein sequence space through computational evolution. Fluorescence intensity is predicted using an ESM-1b-based oracle landscape simulator (Ren et al., 2022), where character '-' denotes residue deletion. The computational evolution process is illustrated in Figure 5.

We conduct 10 condensing trials and select the top 2 sequences from each trial with the highest fitness while keeping the number of deleted residues no less than 50. Then we use AlphaFold 3 (Abramson et al., 2024) to predict structures of these 20 sequences and align their structures with the wild-type avGFP (PDB ID: 1EMA). Finally, we filter the top 5 sequences (ccGFP1-5) with the smallest RMSD to the wild-type structure, as in Figure 5. As we see, AlphaDE successfully evolves the truncated, non-folding sequence to predictedly functional, folded proteins with fewer residues than the wild-type avGFP. More details, including the comparison of AlphaDE without the fine-tuning step and the amino acid residues of ccGFP1-5's protein sequences, can be referred to in Appendix K.

## 5 CONCLUSION

In this work, we propose AlphaDE for the challenging protein directed evolution problem. AlphaDE consists of the fine-tuning step and the MCTS test-time inference step based on the protein language models to attain superior evolution ability. Benchmark experiments on eight tasks show that AlphaDE achieves state-of-the-art performance compared with various baselines, even in a few-shot setting. Additionally, we also demonstrate that AlphaDE could be utilized to computationally condense the protein sequence space.

For future work, on the one hand, integrating large natural language models into AlphaDE to provide the explainability of the evolution process is interesting. On the other hand, applying AlphaDE to industrial applications, such as improving the activity of enzymes, has great economic prospects.

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

## A  DESCRIPTIONS OF BENCHMARK TASKS

Here we briefly describe the eight benchmark tasks constructed by Ren et al. (Ren et al., 2022).

- Green Fluorescent Proteins (avGFP). Green Fluorescent Proteins from *Aequorea victoria*, which can exhibit bright green fluorescence when exposed to light in the blue to the ultraviolet range, are used as biosensors to detect gene expressions and protein locations. Here, we optimize the wild-type sequence in the search space with a size of $20^{238}$ to get higher fluorescence intensity.

- Adeno-associated Viruses (AAV). AAVs are a group of small viruses belonging to the family of dependoviruses, which show great potential in the field of gene therapy. Here we optimize a 28-amino acid segment (position 561-588) of the VP1 protein located in the capsid of the Adeno-associated virus to design more capable sequences measured by AAV liabilities.

- TEM-1 $\beta$-Lactamase (TEM). TEM-1 $\beta$-Lactamase protein resisting penicillin antibiotics in E.coli is widely studied to understand the mutational effect and fitness landscape (Bershtein et al., 2006; Jacquier et al., 2013). Here we optimize the thermodynamic stability in the protein sequence space with a size of $20^{286}$.

- Ubiquitination Factor Ube4b (E4B). Ubiquitination factor Ube4b plays an important role in the trash degradation process in the cell by interacting with ubiquitin and other proteins. We focus on designing E4B with higher enzyme activity. The size of the search space is $20^{102}$.

- Aliphatic Amide Hydrolase (AMIE). Amidase encoded by amiE is an industrially-relevant enzyme from *Pseudomonas aeruginosa*. We seek to optimize amidase sequences that lead to great enzyme activities, which defines a search space with $20^{341}$ sequences.

- Levoglucosan Kinase (LGK). Levoglucosan kinase converts LGK to the glycolytic intermediate glucose-6-phosphate in an ATP-dependent reaction. Here we optimize in a protein sequence space of $20^{439}$ for improving enzyme activity fitness.

- Poly(A)-binding Protein (PAB1). PAB1 functions by binding to multiple adenosine monophosphates (poly-A) using the RNA recognition motif. We optimize to improve binding fitness. The search space size is $20^{75}$ on a segment of the wild-type sequence.

- SUMO E2 conjugase (UBE2I). Using human SUMO E2 conjugase to map the functions of human genomes is significant for scientific research. We improve the fitness measured by growth rescue rate at high temperature in a yeast strain with a search space sized $20^{159}$.

## B  DETAILS OF HOMOLOGOUS SEQUENCE DATASETS

Here, we describe the homologous sequence datasets of the eight benchmark tasks, which are used in the fine-tuning step of AlphaDE. Their fitness distributions and total numbers of sequences are plotted in Figure 6. Please note that we do not use these sequences' fitness labels in either the pretrain, fine-tune, or MCTS inference steps of AlphaDE. We use these protein sequences only for the unsupervised masked language modeling learning in the fine-tuning step. At the same time, we show AlphaDE works with homologous sequences retrieved from the homology search in Appendix J.

## C  HYPERPARAMETER SETTINGS OF ALPHADE

Here we list the hyperparameters in the fine-tuning step and MCTS inference step of AlphaDE. As there are many hyperparameters in both the protein language model and MCTS, we always try to follow the common practice of hyperparameters in previous works to mitigate the effort to tune hyperparameters. We found the hyperparameter settings of AlphaDE work well across tasks without specific tuning for each task.

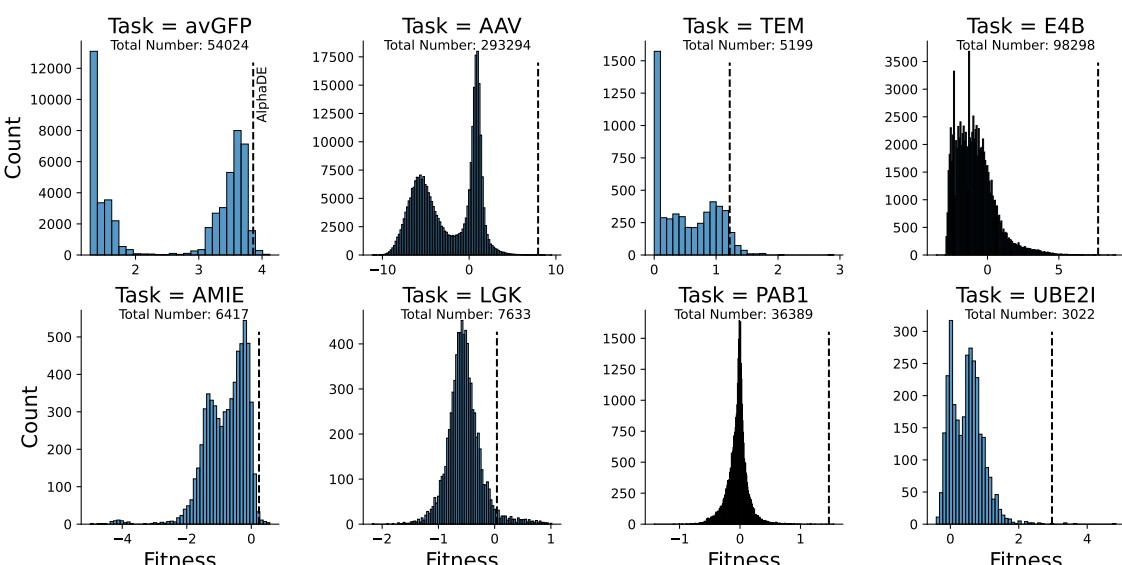

Figure 6: The sequence fitness distributions of each protein dataset used in each task. The black dashed line indicates the fitness location of the best sequences generated by AlphaDE.

### C.1 HYPERPARAMETERS OF FINE-TUNING

For the hyperparameters of fine-tuning, we follow the default setting in the fine-tuning script given by the Transformers package (the most widely used Python package for developing large language models). The fine-tuning script using a masked language modeling loss is referred to GitHub. We fine-tune the pretrained protein language models on each task's homologous sequence dataset for 3 epochs and the batch size per device is set to 8. The learning rate is $5 \times 10^{-5}$ and the optimizer is AdamW. The learning rate scheduler type to use is linear while the weight decay is not enabled. These hyperparameter settings in the fine-tuning step are the same for the ESM2-series, ProtBert, and ESM-1b models. These fine-tuning hyperparameters (learning rate, optimizer, epochs, and so on) are default configurations in large language models and work well in our setting. We only set the batch size per device to 8 for low GPU memory cost. The GPU memory should be slightly larger than 32GB to ensure the fine-tuning on the LGK task with the longest protein sequence, with the protein language model size of 650M. We do not fine-tune the ESM2-3B model and the ESM2-15B model, as fine-tuning the two models requires GPUs with much larger memory. At the same time, the results from Table 3 indicate that the small fine-tuned ESM2-35M model owns enough evolution ability to help AlphaDE evolve proteins.

### C.2 HYPERPARAMETERS OF MCTS

For the hyperparameters of the MCTS inference step in AlphaDE, we follow the default setting in EvoPlay. The constant $c$ in Eq. (3) is a trade-off coefficient to balance exploitation and exploration and is set to 10 for all tasks. The number of simulation rollout times in the Rollout step is set to 200. For an episode, there are several termination conditions. First, AlphaDE terminates when it meets the maximum tree depth (i.e., maximum mutation number), which is set at 100. Second, if the current move is invalid when the state sequence remains unchanged or changes to a previously generated sequence, the current episode terminates. The third termination condition is that the current mutated sequence's fitness value is smaller than the fitness value

of the sequence it mutates from. We also test the sensitivity of key hyperparameters of MCTS, including exploration constant, tree depth, and rollout number, in Appendix H.

For the training of the value network, the update is performed after each episode. The number of training steps for each update is 5, while the batch size for each training step is set to 32. The size of the replay buffer while stores experiences is set to 10000. Here, the optimizer is Adam and the learning rate is $2 \times 10^{-3}$. The weight decay is $1 \times 10^{-4}$. The loss is calculated based on the mean squared error between the predicted fitness value by the value network and the simulated fitness value by the oracle model. These hyperparameter settings in the MCTS inference step are the same for the AlphaDE with ESM2-series, ProtBert, and ESM-1b models. For the MCTS inference step, the GPU memory of less than 8 GB is enough for all tasks.

### C.3 VALUE NETWORK SETTINGS IN ALPHADE

The value network of AlphaDE is with the same network architecture as in EvoPlay (Wang et al., 2023), which consists of 4 convolutional neural network layers and 2 dense layers. As the same network architecture works well in AlphaDE, we do not modify the value network architecture. The only difference is that we do not specify the activation function in the output layer, as we found that the predicted fitness value of different tasks has different value ranges, and we do not restrict the output range of the value network.

## D ALPHADE WITH DIFFERENT EMS2 MODEL SIZES

We also provide the performance of AlphaDE with other ESM2-series models of different sizes such as 8M, 150M, and 650M in Table 3. We find no significant differences between different ESM2 versions while ESM2-35M performs stably, indicating that the evolution information encoded in the fine-tuned ESM2-35M is enough for AlphaDE to conduct an effective tree search for directed evolution in tested protein tasks.

Table 3: Comparison with different ESM2-series model sizes. We present the maximum fitness scores obtained in 1000 black-box oracle queries. Results are averaged over five independent trials.

| AlphaDE Model | avGFP | AAV | TEM | E4B | AMIE | LGK | PAB1 | UBE2I |
|---|---|---|---|---|---|---|---|---|
| ESM2-8M | 3.86 | 8.30 | 1.21 | 7.86 | 0.24 | 0.02 | 1.14 | 2.90 |
| ESM2-35M | 3.86 | 7.95 | 1.22 | 7.75 | 0.24 | 0.04 | 1.47 | 2.97 |
| ESM2-150M | 3.86 | 7.96 | 1.22 | 7.68 | 0.25 | 0.04 | 1.64 | 2.89 |
| ESM2-650M | 3.86 | 8.15 | 1.22 | 7.67 | 0.25 | 0.04 | 1.08 | 2.84 |

## E FITNESS ORACLES

The use of TAPE and ESM-1b oracles follows the standard evaluation procedure in previous works (Ren et al., 2022; Wang et al., 2023; Qiu et al., 2024). The ESM-1b oracle is used in the avGFP condensing experiment as it supports some special characters, including the token '-'. We regard the character token '-' as an amino acid residue deletion and utilize ESM-1b to predict the fitness of the condensed avGFP sequences. These oracle models and their download script could be obtained from GitHub.

## F ABLATION STUDY OF ALPHADE

In this section, we conduct the ablation study to validate each component of AlphaDE. Specifically, EvoPlay is an advanced MCTS framework for directed evolution without the protein language model as mutation guidance, where an actor network is learned from scratch. Therefore, to quantify the contribution of the

pretrained protein language model for mutation action guidance, we compare AlphaDE (MCTS that uses pretrained ESM2) in Section 4.3 with EvoPlay (MCTS that uses an actor from scratch). We see AlphaDE with pretrained ESM2 already outperforms EvoPlay in most tasks. To quantify the contribution of the fine-tuned protein language model, we compare the results of AlphaDE with fine-tuned ESM2 in Table 1 and AlphaDE with pretrained ESM2 in Figure 4. We extract these results to include in Table 4 below for a direct comparison. It is clear that the fine-tuning step in AlphaDE contributes much to the performance.

Table 4: Ablation study of the fine-tuning step in AlphaDE. We present the maximum fitness scores obtained in 1000 black-box oracle queries. Results are averaged over five independent trials.

| Ablation of fine-tuning step | avGFP | AAV | TEM | E4B | AMIE | LGK | PAB1 | UBE2I |
|---|---|---|---|---|---|---|---|---|
| MCTS with fine-tuned ESM2-35M model (AlphaDE) | **3.86** | **7.95** | **1.22** | **7.75** | **0.24** | **0.04** | **1.47** | **2.97** |
| MCTS with pretrained ESM2-35M model (AlphaDE) | 3.13 | -0.72 | 0.89 | 0.38 | -7.08 | -0.01 | 0.82 | 1.70 |
| MCTS with actor network learning from scratch (EvoPlay) | 1.72 | -3.45 | 0.01 | -0.40 | -0.88 | -1.09 | 0.34 | 1.87 |

At the same time, we also conduct the ablation of the whole MCTS search part with greedy search and beam search. The results are shown in Table 5, which highlights the contribution of the MCTS search in AlphaDE.

Table 5: Ablation study of the MCTS inference step in AlphaDE. We present the maximum fitness scores obtained in 1000 black-box oracle queries. Results are averaged over five independent trials.

| Ablation of MCTS | avGFP | UBE2I | E4B | PAB1 |
|---|---|---|---|---|
| AlphaDE (MCTS with finetuned ESM2-35M model) | **3.865** | **2.975** | **7.746** | **1.469** |
| Greedy search with finetuned ESM2-35M model | 3.863 | 2.967 | 1.112 | 0.216 |
| Beam search with finetuned ESM2-35M model | 3.813 | 1.310 | 5.247 | 0.207 |

## G  BENCHMARK EXPERIMENTS WITH ESM-1B ORACLE

We conduct the benchmark experiments with ESM-1b oracle as Ren et al. (2022). Results are shown in Table 6. AlphaDE again beats various competitive baselines in the benchmark setting of EMS-1b oracle.

Table 6: Benchmark experiments with another ESM-1b oracle. We present the maximum fitness scores obtained in 1000 black-box oracle queries. Results are averaged over five independent trials.

| Task | AlphaDE | MLDE | TreeNeuralTS | TreeNeuralUCB | EvoPlay | PEX |
|---|---|---|---|---|---|---|
| avGFP | **3.991** | 2.443 | 3.707 | 3.785 | 3.835 | 3.754 |
| UBE2I | **3.989** | 1.434 | 3.632 | 3.653 | 0.647 | 3.101 |

## H  HYPERPARAMETER SENSITIVITY

We study the sensitivity of several important hyperparameters in AlphaDE's MCTS part, including the exploration constant, tree depth, and rollout number. Results are averaged over five independent trials. We use the top 1000 sequences because the results of the top 1 are good enough to be less discriminative.

### H.1 HYPERPARAMETER SENSITIVITY OF EXPLORATION CONSTANT

In Equation 3, the constant $c$ balances the exploration and exploitation. Here we conduct a hyperparameter sensitivity study to investigate how $c$ influences the performance of AlphaDE. We run AlphaDE with different $c$ values such as 0.1, 1, 10, 100, and 1000 in the avGFP and UBE2I tasks. The results are summarized in Table 7. We see that if $c$ is too large such as 100 and 1000, AlphaDE's performance decreases significantly. Otherwise, AlphaDE achieves similar performance when $c = 0.1$, 1, and 10. In this paper, we set $c = 10$ as a default value and do not specifically tune $c$ for each task. However, we should consider that the best $c$ value may differ in tasks and require further investigation when running AlphaDE on a specific task.

Table 7: Sensitivity study of the exploration constant $c$. We present the average fitness scores obtained in 1000 black-box oracle queries. Results are averaged over five independent trials.

| $c$ | 0.1 | 1 | 10 (default) | 100 | 1000 |
|---|---|---|---|---|---|
| avGFP | 3.72 | 3.73 | 3.72 | 3.59 | 3.49 |
| UBE2I | 2.60 | 2.45 | 2.49 | 2.03 | 1.06 |

### H.2 HYPERPARAMETER SENSITIVITY OF TREE DEPTH

For the tree depth, we give the sensitivity study results in Table 8. We see that the tree depth does not affect AlphaDE's performance much, which indicates that AlphaDE is robust to the tree depth.

Table 8: Sensitivity study of the tree depth. We present the average fitness scores obtained in 1000 black-box oracle queries. Results are averaged over five independent trials.

| tree depth | 1 | 5 | 10 | 100 (default) | 1000 |
|---|---|---|---|---|---|
| avGFP | 3.71 | 3.73 | 3.70 | 3.72 | 3.70 |
| UBE2I | 2.46 | 2.51 | 2.45 | 2.49 | 2.67 |

### H.3 HYPERPARAMETER SENSITIVITY OF ROLLOUT NUMBER

For the rollout number, we give the sensitivity study results in Table 9. We see that the rollout number does not affect AlphaDE's performance much, which indicates that AlphaDE is robust to the rollout number. Meanwhile, we also notice that at the extreme setting where the rollout number is 1 in the task of avGFP, the performance of AlphaDE decreases significantly, which necessitates the importance of multiple rollouts at the reached leaf node.

Table 9: Sensitivity study of the rollout number. We present the average fitness scores obtained in 1000 black-box oracle queries. Results are averaged over five independent trials.

| tree depth | 1 | 10 | 50 | 100 | 200 (default) | 1000 |
|---|---|---|---|---|---|---|
| avGFP | 2.26 | 3.70 | 3.70 | 3.72 | 3.72 | 3.71 |
| UBE2I | 2.43 | 2.42 | 2.35 | 2.49 | 2.49 | 2.41 |

## I DIVERSITY OF EVOLVED SEQUENCES BY ALPHADE

Here, we give the diversity of the evolved sequences by AlphaDE. The diversity is calculated by the top $K$ sequences from each trial and there are 5 trials for each task. The top sequences are ranked by the fitness

value. Then diversity equals to the number of unique sequences divided by $5 \times K$. 100% indicates all the generated sequences are different. $K$ is set at 1, 10, 100, and 1000 respectively and the results are shown in Table 10. We see that AlphaDE generates diverse protein sequences while the top 1 sequences from each trial are different.

Table 10: Diversity of generated sequences by AlphaDE. The diversity is calculated with sequences collected from 5 trials. The top sequences are ranked according to their fitness values.

| Top | avGFP | AAV | TEM | E4B | AMIE | LGK | PAB1 | UBE2I |
|-----|-------|-----|-----|-----|------|-----|------|-------|
| 1 | 100.00% | 100.00% | 100.00% | 100.00% | 100.00% | 100.00% | 100.00% | 100.00% |
| 10 | 100.00% | 100.00% | 100.00% | 100.00% | 100.00% | 100.00% | 100.00% | 100.00% |
| 100 | 99.60% | 96.80% | 100.00% | 100.00% | 100.00% | 100.00% | 100.00% | 100.00% |
| 1000 | 96.96% | 93.80% | 99.14% | 98.96% | 99.12% | 93.22% | 98.82% | 99.64% |

On the other hand, we are also interested in the repetition rate that how often the generated sequences of AlphaDE exist in the task fine-tuning dataset. The repetition rate is calculated by the top $K$ sequences from each trial and there are 5 trials for each task. The top sequences are ranked by the fitness value. Then repetition rate equals to the number of repeated sequences between the top $K$ sequences and the fine-tuning dataset divided by $5 \times K$. $K$ is set at 1, 10, 100, and 1000 respectively and the results are shown in Table 11. From Table 11, we see that, in most tasks, the repetition rate maintains at a very low level. This indicates that AlphaDE generates novel sequences instead of repeating the sequences from the fine-tuning dataset. We also note that, as an exception, for the top 1 sequences of task AAV, the repetition rate is relatively high. But this high repetition rate drops as $K$ increases. For example, the repetition rate of the top 10 AAV sequences drops to 26%, which means the high-fitness sequences are mostly different from the fine-tuning sequences.

Table 11: Repetition rate of generated sequences by AlphaDE. The repetition rate is calculated with sequences collected from 5 trials. The top sequences are ranked according to their fitness values.

| Top | avGFP | AAV | TEM | E4B | AMIE | LGK | PAB1 | UBE2I |
|-----|-------|-----|-----|-----|------|-----|------|-------|
| 1 | 0.00% | 80.00% | 0.00% | 0.00% | 0.00% | 0.00% | 0.00% | 0.00% |
| 10 | 0.00% | 26.00% | 0.00% | 2.00% | 0.00% | 0.00% | 0.00% | 0.00% |
| 100 | 0.00% | 8.60% | 1.20% | 0.40% | 0.00% | 0.00% | 0.00% | 0.00% |
| 1000 | 0.62% | 7.64% | 3.80% | 3.52% | 0.82% | 0.00% | 1.16% | 1.66% |

## J  FINE-TUNING WITH SEQUENCES FROM HOMOLOGY SEARCHING

In the benchmark experiments, almost all the fine-tuned sequence datasets are from deep mutational scanning (DMS), which are not always available for most of the proteins. Therefore, in this section, we introduce the homology searching technique to construct the dataset of homologous sequences for the fine-tuning step in AlphaDE. Next, we use avGFP as an example. Specifically, we use the phmmer homology searching procedure of the biosequence analysis tool HMMER (Potter et al., 2018) to find homologous sequences of the starting weakest sequence in the avGFP task. We use the default settings of phmmer to search the SwissProt database, UniProt database, Reference Proteomes database, and PDB database. After filtering sequences with the same length as avGFP and removing three duplicate sequences in the DMS dataset, we found 236 unique sequences to construct the avGFP phmmer dataset, which has no overlap with the avGFP DMS dataset. Then we follow the standard AlphaDE fine-tuning step on the avGFP phmmer dataset and perform the MCTS step with the fine-tuned model (here $c$ is set at 1.0 for a better performance). Results are shown in Table 12 below. We see that, AlphaDE, which uses the phmmer homologous sequence dataset for fine-tuning, also achieves superior performance.

Table 12: Results of different fine-tuning datasets. We present the maximum fitness scores obtained in 1000 black-box oracle queries. AlphaDE (phmmer) is also averaged over five independent trials.

| Method | AlphaDE (DMS) | AlphaDE (phmmer) | PEX | AdaLead | TreeNeuralTS |
|---|---|---|---|---|---|
| avGFP | 3.86 | 3.83 | 2.97 | 2.61 | 2.44 |

## K DETAILS OF CONDENSING AVGFP

When initializing the deleted avGFP sequences, we keep the $\beta$-barrel residues and the chromophore related residues. The $\beta$-barrel residues involve the residue 1 to residue 38. The key chromophore residues are residue 65, 66, and 67 (Hayes et al., 2025), and we set the chromophore-related residues to be residue 58 to residue 74. Then we delete half of the left residues, which results in a starting deleted sequence with length 146. In contrast, the wild-type sequence has a length of 238 residues. The wild-type avGFP sequence, the starting deleted sequence, and the final filtered ccGFP1-5 sequences are given in Table 13. When filtering, the amino acid sequences of ccGFP1-5 are fed into AlphaFold 3 server to predict their structures. Then we align these structures with the wild-type structure (PDB ID: 1EMA) by PyMol to calculate the RMSD distances. Although this is a computational proof-of-concept task, it shows the great potential of AlphaDE for different purposes with directed evolution.

Table 13: Amino acid sequences of avGFP variants in the protein sequence condensing experiment.

| Name | Amino Acid Sequence |
|---|---|
| wild-type | MSKGEELFTGVVPILVELDGDVNGHKFSVSGEGEGDATYGKLTLKFICTTGKLPVPWPTLVTTLSYGVQCFSRYPDHMK QHDFFKSAMPEGYVQERTIFFKDDGNYKTRAEVKFEGDTLVNRIELKGIDFKEDGNILGHKLEYNYNSHNVYIMADKQK NGIKVNFKIRHNIEDGSVQLADHYQQNTPIGDGPVLLPDNHYLSTQSALSKDPNEKRDHMVLLEFVTAAGITHGMDELYK |
| starting | MSKGEELFTGVVPILVELDGDVNGHKFSVSGEGEGDATGLLFCTKPPPTLVTTLSYGVQCFSRYDMQDFSMEYQRIFDGY TAVFGTVREKIFEGIGKENNHVIAKKGKNKRNEGVLDYQTIDPLPNYSQASDNKDMLEVAGTGDLK |
| ccGFP1 | MSKGEELFTLVVPILVELRGDVNGHKFSVSGEGEGNATGLTLKFCTTGKLPVPWPTLVTTLSYGVQCFSRYDVMQHDFK SAMEGYVQRTIFFDGYTRAEVFGDTVRELKGIFEGIGKENNSHNVWIADKKGIKNFKRNEGSVVADHYQTFIDPVLPNILS TQSASDNKRDHMILLEGVAGHHGMDLYK |
| ccGFP2 | MSKGEELFTLVVPILVELRGDVNGHKFSVTGEGEGNATGLTLKFCTTGKLPVPWPTLVTTLSYGVQCFSRYDVMQHDFK SAMEGYVQRTIFFDGYTRAEVFGDTVRELKGIFEGIGKENNSHNVWIADKKGIKNFKRNEGSVVADHYQTFIDPVLPNILS TQSASDNKRDHMILLEGVAGHHGMDLYK |
| ccGFP3 | MSKGEELFTGVVPILVELDGDVNGHKFSVSGEGEGDATGQLTLKFCTTKLPVAWPTLVTTPSYGVQCFSRYDMKQHDQS AMEYAQRDIFFKDYTRAVKFGTLVRELKVIDFEGNILGKEYNNSHVIADKQKGIKNKRNEGVQLDHYQQNTPIVDPVLLP LNHYSQSALSDNEKRDMLLEFVTAGTGLK |
| ccGFP4 | MSKGIELFTGVPILVELDGDVNGHKFSVSGEGEGDASGKLLFCTTKPVPCTLVTTLSYGVQCFSRYPDMKQHDFKSAMR YQRRTIFDGNYTAVFGTLVRIELKGIDFKEGIGKENYNSHVIMADKQKNGIKVNFKRHTIEGVLADHYQTIDGPVLPNHYL STQASIDNKDMVLEVTAAGTHGMDLK |
| ccGFP5 | MSKGIELFTGVPILVELDGDVNGHKFSVSGEGEGDASGKLLFCTTKPVPCTLVTTLSYGVQCFSRYPDMKQHDFKSAMR YQRRTIFDGNYTAVFGTLVRIELKGIDFKEGIGKENYNSHVIMADKQKNGKVNFKRHTIEGVLADHYQTIDGPVLPNHYL STQASIDNKDMVLEVTAAGTHGMDLK |

At the same time, we also compare with AlphaDE with pretrained protein language model to validate whether it recovers the folded structure of wild-type avGFP. We follow the same pipeline as condensing avGFP with standard AlphaDE. The comparison of the AlphaDE with pretrained ESM2-35M and standard AlphaDE is shown in Table 14 below. It is clear that the condensed avGFPs by AlphaDE with pretrained ESM2-35M have much larger RMSDs, indicating the recovered folded structures are less similar to the wild-type avGFP structure than the standard AlphaDE.

Table 14: Comparison of AlphaDE with pretrained ESM2-35M and standard AlphaDE with fine-tuned ESM2-35M for condensing avGFP. The metric is RMSD to the wild-type avGFP (PDB ID: 1EMA).

| RMSD ($\downarrow$) | 1 | 2 | 3 | 4 | 5 |
|---|---|---|---|---|---|
| standard AlphaDE with fine-tuned ESM2-35M | 1.10 | 1.40 | 1.70 | 1.83 | 1.92 |
| AlphaDE with pretrained ESM2-35M | 5.92 | 6.27 | 6.79 | 7.55 | 8.08 |

## L    COMPUTATIONAL EFFICACY OF ALPHADE

Here we give the running time of the fine-tuning step and the MCTS inference step. The protein language model here is ESM2-35M, which is the default configuration in AlphaDE. The running time is shown in Table 15. For the fine-tuning step, the running time mainly depends on the number of sequences in the dataset. We use three NVIDIA GPUs for this fine-tuning step and one NVIDIA GPU for the MCTS inference step. When fine-tuning, the GPU memory cost depends on the size of protein language models and the length of protein amino acid sequences.

Table 15: Running time of AlphaDE with ESM2-35M. The unit of running time is the hour.

| Step | avGFP | AAV | TEM | E4B | AMIE | LGK | PAB1 | UBE2I |
|---|---|---|---|---|---|---|---|---|
| Fine-Tuning | 0.38 | 1.22 | 0.04 | 0.45 | 0.06 | 0.10 | 0.17 | 0.02 |
| MCTS Inference | 4.36 | 1.66 | 4.46 | 3.04 | 3.93 | 1.85 | 1.12 | 1.08 |

Meanwhile, we additionally provide the computational cost of competitive baselines such as TreeNeuralTS and TreeNeuralUCB for a comparison in the task avGFP. We also compare with EvoPlay. The results are included in Table 16. As AlphaDE utilizes the protein language model, it is expected to take a longer time to run. We also see that, the running time of AlphaDE is acceptable, compared with other baselines.

Table 16: Running time of AlphaDE and different baselines in the task of avGFP. The unit of running time is the hour. The resulting running hour values are averaged over 5 trials.

| Task | AlphaDE | TreeNeuralTS | TreeNeuralUCB | EvoPlay |
|---|---|---|---|---|
| avGFP | 4.74 | 2.33 | 1.79 | 0.69 |

## M    LIMITATION

Our study has limitations under extensive consideration. First, the oracle models for fitness evaluation may have biases and cannot replace the real-world wet-experiment measurements. Second, the fine-tuning step requires homologous sequences, which may not always exist for a specific protein. If there are novel or poorly characterized proteins with seldom homologous sequences, the application of our method may be restricted. Luckily, AlphaDE supports the few-shot fine-tuning as indicated in Section 4.2, which greatly reduces the needed number of homologous sequences. Additionally, as in Section 4.3, our method supports zero-shot mode with the pretrained protein language models if homologous sequences are not available.

## N    BROADER IMPACTS

Directed evolution is a powerful computational tool for protein engineering. The proposed AlphaDE significantly boosts the efficiency of in-silicon directed evolution, which exhibits great potential for real-world

protein engineering applications. At the same time, we emphasize safety concerns that it can be misused to generate pathogenic mutations and harmful bio-agents. Hence, we declare that AlphaDE should be restricted to research purposes, and any applications should undergo comprehensive experiments and human inspections.

