# OpenReview forum: "Boosting In-Silicon Directed Evolution with Fine-Tuned Protein Language Model and Tree Search"
_ICLR.cc/2026/Conference — ICLR 2026 Conference Withdrawn Submission_

### Official Review · Reviewer_KJoU · 2025-10-28

**Soundness:** 2
**Presentation:** 1
**Contribution:** 2
**Rating:** 2
**Confidence:** 3

**Summary:**

The authors claim to contribute:
- a new framework for doing in silico directed evolution which takes advantage of pre-trained protein language models as large evolutionary priors.
- a test-time inference procedure using MCTS on top of a fine-tuned PLM to further optimize protein function.
- present improved in silico directed evolution of protein compared to a variety of design methods on a variety of assay-based protein predictors.

**Strengths:**

- Idea to bring test-time inference like MCTS from LLMs to protein language models seems novel (I’m not super sure though, not super well contextualized).
- Results (Table 1) look substantially better than other methods on most of the problems. I’m wondering if it’s a fair comparison given this, some almost seem to good to be true. I don’t personally understand why I would expect such substantial gains over various baseline methods, and over zero-shot.
- Experiments compare to many methods for protein design, though some relevant ones are missing.
- Ablations and hyperparameters are documented pretty rigorously in the appendix.

**Weaknesses:**

- Poorly motivated — other work exists using pre-trained PLMs for protein design / directed evolution; these aren’t even cited (e.g., [1], [2], [3], [7]). Methods specifically for multi-round design, like [6], aren’t considered.
- It's not really made clear why anything about AlphaDE is specific or particular to directed evolution (which is iterative) and not just general protein design.
- The background section, which appears to serve as a related works section, lists some prior works that can be used to do in silico protein design. However, the authors don’t explain why their method might be a better idea than any of these. They only cite wanting to use PLMs for directed evolution, and don’t motivate this convincingly. Only the last sentence of the background explains the difference from previous works. It also doesn't explain why fine-tuning without using any supervised data is a desirable strategy, when labels are available for all of the test cases they consider in experiments (Table 1). Wouldn't one expect using labels to be helpful?
- Experiments don't compare to other methods using PLMs, like [1], [2], and [5].
- It looks like for their experiments (Table 1), they chose to use PEX instead of the stronger PEX+MuFacNet without explanation, even though both are shown in the plots where they take their baseline results directly from [4]. For avGFP, AAV, and TEM problems, PEX+MuFacNet performs better than PEX but this isn't reported.
- Fig. 2 doesn’t show how good uniform sampling is as baseline, or the base pre-trained model. Even after 1 fine-tuning epoch, AlphaDE looks pretty good, so one might expect the base model to also be pretty good. It's also unclear whether AlphaDE is generalizing substantially, or just sampling from sequences in the fine-tuning dataset--their fitness distribution should also be shown.
- It’s not surprising that given avGFP’s chromophore (Sec 4.4 / Fig. 5), a PLM can generate a protein very similar to the wild-type. These models have been trained on avGFP. I’d want a comparison to without fine-tuning, or without MCTS to be convinced that their AlphaDE pipeline contributes to this.

I would recommend rejection primarily because their method is not well motivated and situated within related works. Other work exists using pre-trained PLMs for protein design and directed evolution; these aren’t even cited (e.g., [1], [2], [3], [5], [7]). Methods specifically for multi-round design, like [6], aren’t considered. The authors don’t provide a convincing explanation as to why one should expect fine-tuning a PLM on sequences found from a simple homology search, and then using MCTS would outperform all these other design methods. A PLM-based prior could be easily integrated into many of the baseline methods (e.g., CbAS, CMA-ES, BO, probably others that I’m less aware of) for a more equal comparison, but this doesn’t seem to have been done. If the takeaway is primarily that using a PLM as a prior is useful, then the paper should have been written differently. There are no error bars for any of the tables although they claimed to be over replicates. Plots showing the mean value of sequences sampled by each method at each iteration would be more transparent but these are not shown.

More clearly contextualizing your method as motivated by previous works would help it be much more understandable. Currently, it reads as though you’ve presented a method that works quite well but the reasons aren’t clear, and a reader is forced to take your word for it and just try it out. Adding more intuition throughout the paper for why your method should be better than alternative approaches could help remedy this.


Citations:

[1] Jiang, Kaiyi, et al. "Rapid in silico directed evolution by a protein language model with EVOLVEpro." _Science_ 387.6732 (2024): eadr6006.

[2] Yang, Jason, et al. "Steering Generative Models with Experimental Data for Protein Fitness Optimization." arXiv preprint arXiv:2505.15093 (2025).

[3] Tran, Thanh VT, and Truong Son Hy. "Protein design by directed evolution guided by large language models." IEEE Transactions on Evolutionary Computation (2024).

[4] Ren, Zhizhou, et al. "Proximal exploration for model-guided protein sequence design." International Conference on Machine Learning. PMLR, 2022.

[5] Nisonoff, Hunter, et al. "Unlocking guidance for discrete state-space diffusion and flow models." ICLR (2025).

[6] Yang, Jason, et al. "Active learning-assisted directed evolution." _Nature Communications_ 16.1 (2025): 714.

[7] Wang, Chenyu, et al. "Fine-tuning discrete diffusion models via reward optimization with applications to dna and protein design." ICLR (2025).

**Questions:**

What are the fitness values for the homologous sequences that the PLM is fine-tuned for each protein dataset? What part of your method exactly is yielding a substantial gain—it appears to be the fine-tuning from Table 2, but that doesn’t make sense given that you wrote that the homologous sequences are generated from the worst sequence in the dataset. If the homologous sequences fine-tuned on really are low-value, why is it so helpful to fine-tune on them? Would it similarly help to just fine-tune on the 100 worst sequences in the dataset?
How are the baseline methods initialized? Completely randomly (which may be unfair), or with samples from a zero-shot version of the PLM used for AlphaDE? It seems like even without fine-tuning, the PLM has a max-value greater than or similar to many of the baselines, which doesn’t really seem fair.

---

> ### Author Response · Authors · 2025-11-15
> **Rebuttal part I**
>
> We sincerely appreciate Reviewer KJoU for the detailed comments. We provide the following clarifications.
>
> 1.	Poorly motivated — other work exists using pre-trained PLMs for protein design / directed evolution; these aren’t even cited (e.g., [1], [2], [3], [7]). Methods specifically for multi-round design, like [6], aren’t considered.
>
> Regarding the relevant works [1, 2, 3, 5, 6, 7], we will address them in the revised manuscript as follows:
>
> For works [1] and [6]: We have added the following citations - “For example, [1] and [6] present the multi-round active learning to rapidly improve protein activity, which alternates between collecting sequence fitness using a wet-lab assay and training a protein language model to prioritize new sequences to screen in the next round.” As they use the dry-wet loop on real-world cases, we cannot directly compare with them. However, their core optimization techniques ([1] uses greedy selection and [6] uses BO) are already benchmarked in our paper.
>
> Fow work [2]: We have added the following citations – “At the same time, some works \cite{widatalla_aligning_2024, [2]} utilize labeled sequence-fitness pairs to steer protein language models for de novo protein sequence design to generate novel and high-quality sequences, showing the superior alignment ability of protein language models.”
>
> For work [3]: We have added the following citations – “Moreover, MLDE [3] introduces an optimization pipeline that utilizes protein language models to pinpoint the mutation hotspots and then suggest replacements by heuristically calculating a k-mer's relevancy and entropy in a sequence.” At the same time, we benchmark [3] in our experiments.
> | Method | avGFP | AAV | TEM | E4B | AMIE | LGK | PAB1 | UBE2I |
> | :--- | :--- | :--- | :--- | :--- | :--- | :--- | :--- | :--- |
> | AlphaDE | **3.86** | **7.95** | **1.22** | **7.75** | **0.24** | **0.04** | **1.47** | 2.97 |
> | MLDE [3] | 2.04 | -3.38 | 0.08 | 3.84 | -1.38 | -0.19 | 0.92 | 2.75 |
>
> [5] and [7]: After careful consideration, we found these works primarily focus on guiding inverse folding models (e.g., ProteinMPNN and FMIF) for de novo stable sequence design, which addresses a problem less relevant to our focus on directed evolution of engineering a given starting sequence. To maintain clarity and focus, we have chosen not to cite them to avoid confusing readers. We are open to further guidance if the reviewer believes this is a misinterpretation.
>
> 2.	It's not really made clear why anything about AlphaDE is specific or particular to directed evolution (which is iterative) and not just general protein design.
>
> We thank the reviewer for this insightful question. The key differentiation lies in the problem formulation. General protein design often aims for de novo generation of novel folds or sequences satisfying global constraints (e.g., stability). In contrast, directed evolution (DE) is inherently an iterative optimization process that starts from a parent sequence and explores a local, functionally relevant mutational landscape. AlphaDE is specifically engineered for this iterative nature of DE. By formulating it as a Markov Decision Process (MDP), our method explicitly models the sequential decision-making of selecting mutations. The integration of AlphaZero-like MCTS is crucial here, as it performs a look-ahead search to simulate and evaluate multi-step mutation paths, effectively planning a trajectory through sequence space. This is fundamentally different from one-shot generation methods. Furthermore, the fine-tuned PLM acts as a task-specific prior, constraining the search to a plausible local neighborhood around the parent sequence, which mirrors the natural evolutionary process of exploring proximal variants. This combination of iterative search with a localized prior makes AlphaDE particularly suited for directed evolution, not just general design.

---

> > ### Author Response · Authors · 2025-11-15
> > **Rebuttal part II**
> >
> > 3.	The background section, which appears to serve as a related works section, lists some prior works that can be used to do in silico protein design. However, the authors don’t explain why their method might be a better idea than any of these. They only cite wanting to use PLMs for directed evolution, and don’t motivate this convincingly. Only the last sentence of the background explains the difference from previous works. It also doesn't explain why fine-tuning without using any supervised data is a desirable strategy, when labels are available for all of the test cases they consider in experiments (Table 1). Wouldn't one expect using labels to be helpful?
> >
> > We have expand our expression from “Different from previous works, in this paper, we explore using the natural large language model technique paradigm to boost in-silicon directed evolution with protein language models.” to “Different from previous works, here we follow a principle of effective searching (i.e., AlphaZero-like MCTS) with strong prior guidance (i.e., fine-tuned protein language model), inspired by the natural large language model technique paradigm such as fine-tuning \cite{shao_deepseekmath_2024} and test-time inference \cite{guan_rstar-math_2025}, to boost in-silicon directed evolution with a novel technique paradigm.”. The superiority of AlphaDE lies in its hybrid architecture: the fine-tuned PLM provides a powerful, localized prior that generic baselines lack, while MCTS performs a more strategic, multi-step search than typical one-shot or myopic optimization methods.
> >
> > 4.	Experiments don't compare to other methods using PLMs, like [1], [2], and [5].
> >
> > As we study the directed evolution problem in our paper, [2] and [5] are less related to our topic, as stated in Q1. For [1], it uses dry-wet loop on real-world cases, we cannot directly compare with it. However, [1] uses greedy selection for optimization, which has already been benchmarked in our paper (Table 5 in Appendix E). At the same time, we conduct the experiments to compare MLDE [3], which introduces an optimization pipeline for in-silicon directed evolution that utilizes protein language models. Results (see Q1) show that AlphaDE performs better than MLDE [3].
> >
> > 5.	It looks like for their experiments (Table 1), they chose to use PEX instead of the stronger PEX+MuFacNet without explanation, even though both are shown in the plots where they take their baseline results directly from [4]. For avGFP, AAV, and TEM problems, PEX+MuFacNet performs better than PEX, but this isn't reported.
> >
> > We appreciate the reviewer's comment. We have now updated Table 1 to include the results for the stronger PEX (MuFacNet) baseline, as suggested. Our reported results for "PEX (CNN)" were intended to distinguish the surrogate model architecture. The updated table now shows that AlphaDE achieves superior performance compared to both PEX variants.
> >
> > | Method | avGFP | AAV | TEM | E4B | AMIE | LGK | PAB1 | UBE2I |
> > | :--- | :--- | :--- | :--- | :--- | :--- | :--- | :--- | :--- |
> > | AlphaDE | **3.86** | **7.95** | **1.22** | **7.75** | **0.24** | **0.04** | **1.47** | 2.97 |
> > | PEX (MuFacNet) | 3.12 | 4.45 | 0.27 | 2.22 | 0.16 | **0.04** | 1.23 | 2.97 |
> > | PEX (CNN) | 2.97 | 2.52 | 0.19 | 2.21 | -0.11 | 0.03 | 1.27 | 2.97 |

---

> ### Author Response · Authors · 2025-11-15
> **Rebuttal part III**
>
> 6.	Fig. 2 doesn’t show how good uniform sampling is as baseline, or the base pre-trained model. Even after 1 fine-tuning epoch, AlphaDE looks pretty good, so one might expect the base model to also be pretty good. It's also unclear whether AlphaDE is generalizing substantially, or just sampling from sequences in the fine-tuning dataset--their fitness distribution should also be shown.
>
> Figure 4 shows the MCTS with the base pre-trained model. Table 4 in Appendix E compares the base model and the fine-tuned model. The fine-tuned model outperforms the base pre-trained model. Table 5 in Appendix E compares the MCTS, greedy search, and beam search. MCTS is more effective than simpler search strategies like beam search or greedy search. Figure 6 in Appendix B shows the fitness distribution.
>
> 7.	It’s not surprising that given avGFP’s chromophore (Sec 4.4 / Fig. 5), a PLM can generate a protein very similar to the wild-type. These models have been trained on avGFP. I’d want a comparison to without fine-tuning, or without MCTS to be convinced that their AlphaDE pipeline contributes to this.
>
> We add this experiment - “At the same time, we also compare with AlphaDE with a pretrained protein language model to validate whether it recovers the folded structure of wild-type avGFP. We follow the same pipeline as condensing avGFP with standard AlphaDE. The comparison of the AlphaDE with pretrained ESM2-35M and standard AlphaDE is shown in the Table below. It is clear that the condensed avGFPs by AlphaDE with pretrained ESM2-35M have much larger RMSDs, indicating the recovered folded structures are less similar to the wild-type avGFP structure than the standard AlphaDE”.
>
> | RMSD (↓) | 1 | 2 | 3 | 4 | 5 |
> | :--- | :--- | :--- | :--- | :--- | :--- |
> | standard AlphaDE with fine-tuned ESM2-35M | 1.10 | 1.40 | 1.70 | 1.83 | 1.92 |
> | AlphaDE with pretrained ESM2-35M | 5.92 | 6.27 | 6.79 | 7.55 | 8.08 |

---

> ### Author Response · Authors · 2025-11-15
> **Rebuttal part IV**
>
> 8.	What are the fitness values for the homologous sequences that the PLM is fine-tuned for each protein dataset? What part of your method exactly is yielding a substantial gain—it appears to be the fine-tuning from Table 2, but that doesn’t make sense given that you wrote that the homologous sequences are generated from the worst sequence in the dataset. If the homologous sequences fine-tuned on really are low-value, why is it so helpful to fine-tune on them? Would it similarly help to just fine-tune on the 100 worst sequences in the dataset? How are the baseline methods initialized? Completely randomly (which may be unfair), or with samples from a zero-shot version of the PLM used for AlphaDE? It seems like even without fine-tuning, the PLM has a max-value greater than or similar to many of the baselines, which doesn’t really seem fair.
>
> Q8.1: What are the fitness values for the homologous sequences that the PLM is fine-tuned for each protein dataset?
>
> A8.1: Figure 6 in Appendix B shows the fitness distribution for each protein dataset.
>
> Q8.2: What part of your method exactly is yielding a substantial gain?
>
> A8.2: Table 4 in Appendix E compares the model from scratch, the pretrained base model and the fine-tuned model. Table 5 in Appendix E compares the MCTS, greedy search, and beam search. We see both the fine-tuned protein language model and MCTS yield a substantial gain.
>
> Q8.3: If the homologous sequences fine-tuned on really are low-value, why is it so helpful to fine-tune on them?
>
> A8.3: The fine-tuning step with homology sequences makes the generated sequences of protein language model to be close to the protein space of avGFP. So, MCTS only explores at the protein space around avGFP, which is efficient for searching.
>
> Q8.4: Would it similarly help to just fine-tune on the 100 worst sequences in the dataset?
>
> A8.4: Yes, pleases see Figure 3 with bottom 20% data with a similar setting. We provide the Max Bottom 20% fitness value (The “Max Bottom 20%” value denotes the maximum fitness value in the bottom 20% data), which is far smaller than the best fitness AlphaDE can achieve.
>
> Q8.5: How are the baseline methods initialized?
>
> A8.5: We strictly follow [1] for the initial setting for baseline methods, including PEX, AdaLead, CbAS, DbAS, DyNA-PPO, BO and CMA-ES. We follow the officially released codebase and configurations of TreeNeuralTS, TreeNeuralUCB, MLDE, and EvoPlay for running while using the TAPE and ESM1b oracles taken from [1].
>
> References
>
> [1] Ren, Zhizhou, et al. "Proximal exploration for model-guided protein sequence design." International Conference on Machine Learning. PMLR, 2022.
>
> Q8.6: Completely randomly (which may be unfair), or with samples from a zero-shot version of the PLM used for AlphaDE? It seems like even without fine-tuning, the PLM has a max-value greater than or similar to many of the baselines, which doesn’t really seem fair.
>
> A8.6: Table 4 in Appendix E compares the model from scratch, the pretrained base model, and the fine-tuned model. Using PLM for searching guidance is our core contribution. It brings a big advantage, and we think it is a promising future direction as PLM encodes rich evolutionary patterns. As we witness PLM shows great potential in protein directed evolution [1-3], so we believe it is a promising future direction.
>
> References
>
> [1] Jiang, Kaiyi, et al. "Rapid in silico directed evolution by a protein language model with EVOLVEpro." Science 387.6732 (2024): eadr6006.
>
> [2] Yang, Jason, et al. "Active learning-assisted directed evolution." Nature Communications 16.1 (2025): 714.
>
> [3] Tran, Thanh VT, and Truong Son Hy. "Protein design by directed evolution guided by large language models." IEEE Transactions on Evolutionary Computation (2024).

---

### Official Review · Reviewer_XhVw · 2025-10-30

**Soundness:** 2
**Presentation:** 3
**Contribution:** 1
**Rating:** 2
**Confidence:** 4

**Summary:**

The work proposes to fine-tune a protein language model to guide the directed evolution for protein design. The work first fine-tune a protein language model with masked language model. Then the fine-tuned model is able to propose mutations. Then the model is used as a policy in an RL framework, where MCTS is performed to find protein sequences with a good fit. From the perspective of machine learning research, the methods in this work are well-known in the machine learning community.

**Strengths:**

The paper is well written and easy to understand. [Though I could not evaluate the experiment results because I am not from the area]

**Weaknesses:**

The most significant issue with the work is the lack of novelty. Most content of the work before the experiment section is known to the field. Specifically, section 3 describes the proposed method, but all the content is known to the community: the problem 3.1 is a well-known problem [1]. The masked language modeling in 3.2 is popularized by BERT [2] and is widely used in network training. The MCTS in 3.3 is popularized by AlphaGo. Therefore, I don't find the innovation in this work.

I could not evaluate the significance of the experiment results as I am not from this area. Even if the results are much better than the state-of-the-art, ICLR might not be the proper venue for this work.

[1] Yang et al. Machine-learning-guided directed evolution for protein engineering. Nature Methods. 2019.
[2] Devlin, Jacob, et al. "Bert: Pre-training of deep bidirectional transformers for language understanding." Proceedings of the 2019 conference of the North American chapter of the association for computational linguistics: human language technologies, volume 1 (long and short papers). 2019.
[3] Silver, David, et al. "Mastering the game of Go with deep neural networks and tree search." nature 529.7587 (2016): 484-489.

**Questions:**

I hope the author could better justify the contribution of leveraging "protein language models into directed evolution for effective exploration".

---

> ### Author Response · Authors · 2025-11-17
> **Rebuttal**
>
> We sincerely appreciate Reviewer XhVw for the detailed comments. We provide the following clarifications.
>
> 1.	I hope the author could better justify the contribution of leveraging "protein language models into directed evolution for effective exploration".
>
> We agree with the reviewer that clearly articulating the contribution is essential. The core contribution of AlphaDE lies not in the individual components of fine-tuning or MCTS in isolation, but in their novel and synergistic integration into a unified framework for directed evolution. While previous works have used protein language models (PLMs) for fitness prediction or employed search algorithms, AlphaDE introduces a distinct paradigm by:
>
> (1) Integrating a Fine-tuned PLM as an Adaptive Evolutionary Prior: we fine-tune it on homologous sequences to create a domain-specific policy. This provides a strong biological prior that guides the search towards functionally plausible regions of sequence space.
>
> (2) Employing Strategic Planning with MCTS: We move beyond simple search strategies (e.g., beam search) by incorporating AlphaZero-style Monte Carlo Tree Search. This enables multi-step lookahead planning, allowing the algorithm to evaluate potential mutation pathways and strategically navigate rugged fitness landscapes by balancing exploration with exploitation.
>
> The key advancement is that these components are not merely used separately but are deeply coupled. The fine-tuned PLM shapes the search policy for MCTS, while MCTS uses this informed policy to conduct a far-sighted exploration that simple methods cannot achieve. This conceptual and algorithmic integration enables a more intelligent and effective exploration of the fitness landscape, which we demonstrate leads to state-of-the-art performance across multiple benchmarks.

---

> > ### Comment · Reviewer_XhVw · 2025-11-20
> > **Thank you for your response**
> >
> > Thank you for your response. I would encourage the author to highlight the technical difficulty in the integration of PLM with MCTS. Does a plain integration work? If so, then it should not be a research paper for a machine learning conference. If not, then the paper may want to show the reason and highlight the solution.

---

> > > ### Author Response · Authors · 2025-11-21
> > > **Rebuttal**
> > >
> > > Dear reviewer XhVw,
> > >
> > > For the assessment of technical difficulty, please refer to Section 3, which gives the method details of AlphaDE, Appendix C for hyperparameter settings of AlphaDE, and Appendix H for hyperparameter sensitivity of AlphaDE. Again, the integration of PLM with MCTS has strong motivation (please see paragraphs 3 and 4 in the Introduction section), and integrating them into the protein directed evolution problem is never a plain integration work.
> > >
> > > At the same time, technical difficulty never determines whether a paper is a research paper for a machine learning conference. For example, do you think integrating a neural network and MCTS to solve Go is not a research paper for a machine learning conference? The motivation, the idea novelty, and the contribution should be among the most important factors for a research paper in a machine learning conference.

---

### Official Review · Reviewer_Z76E · 2025-11-01

**Soundness:** 2
**Presentation:** 2
**Contribution:** 2
**Rating:** 2
**Confidence:** 3

**Summary:**

AlphaDE finetunes PLMs using MLM on homologus protein sequences, then uses this as a policy network to do MCTS towards high-fitness mutations. A value network is also trained online to accelerate rollouts. Evaluation is done on 8 tasks, against other baselines that follow the "fitness landscape exploration" approach to DE tasks.

**Strengths:**

* Low-N fitness prediction is important to enable.
* Formalizing how RL approaches can be used in PLMs is timely and can lead to productive future works.
* On the benchmarks explored, performance looks favorable, e.g. on the TEM task.

**Weaknesses:**

* Though this idea of RL post-training for PLMs holds promise, given the current state of the LLM field, the execution becomes quite important, and I think the paper can do better on this in terms of rigor and following through on failure cases. I personally don't think the idea itself is super novel, and I think what makes a paper like this shine would be to really help readers get intuition on how RL post-training will differ for PLMs. Even in terms of base execution, there are some decisions that don't entirely make sense. For example: why use ESM-1b oracle rather than a more recent model? I get there's a desire for consistency with baselines, but I think it's more important to execute well, and reimplement the baselines if needed.
* Finetuning on homologous sequences is not a new idea; it’s been done since earlier ML for protein design works (Alley et al., 2020, Biswas et al., 2021) as well as recent works (Gordon et al., 2025). This limits the novelty of the work and the completeness of the discussion.
* Computational costs is a lot higher. Appendix L reports that AlphaDE takes 4.74 hours, vs 0.69 hours for EvoPlay.
* The simulated landscapes are very toy settings, limiting its applicability to the real world. This is inherent to fitness landscape exploration type works, and not unique to this paper, but nonetheless limits the ultimate impact.
* Nit: presentation - avoid “impressively” and qualifying words in scientific writing, in line 58.


Alley et al, 2020: https://pubmed.ncbi.nlm.nih.gov/33828272/
Biswas et al., 2021: https://pmc.ncbi.nlm.nih.gov/articles/PMC7067682/
Gordon et al., 2025: https://www.biorxiv.org/content/10.1101/2024.10.03.616542v1

**Questions:**

* IIUC from the Appendix C2, the process stops when the current mutated sequence fitness is lower than the wildtype. Is that overly greedy? If you take out this termination requirement, how often would you see the fitness climb back up? Given how rugged protein landscapes are, this is type of investigation might be interesting.

---

> ### Author Response · Authors · 2025-11-17
> **Rebuttal**
>
> We sincerely appreciate Reviewer Z76E for the detailed comments. We provide the following clarifications.
>
> 1.	Though this idea of RL post-training for PLMs holds promise, given the current state of the LLM field, the execution becomes quite important, and I think the paper can do better on this in terms of rigor and following through on failure cases. I personally don't think the idea itself is super novel, and I think what makes a paper like this shine would be to really help readers get intuition on how RL post-training will differ for PLMs. Even in terms of base execution, there are some decisions that don't entirely make sense. For example: why use ESM-1b oracle rather than a more recent model? I get there's a desire for consistency with baselines, but I think it's more important to execute well, and reimplement the baselines if needed.
>
> To address the concern about how RL post-training differs for PLMs, our paper already contains analysis that we will bring to the forefront. Table 5 (Appendix E), which compares MCTS against greedy and beam search, provides a clear performance differentiation. More importantly, it demonstrates that MCTS is not merely a more efficient sampler but enables a more strategic exploration. By simulating multi-step mutation paths, it can identify high-fitness sequences that require non-obvious, intermediate steps.
>
> Regarding the choice of ESM-1b as an oracle, we prioritized a controlled and fair benchmark. Using the same oracle as established baselines is crucial for an apples-to-apples comparison, ensuring that performance gains are attributable to our search algorithm and not confounded by a more powerful oracle. This is a standard practice for rigorous methodological comparison. At the same time, we also use TAPE as another oracle as shown in Table 1.
>
> 2.	Finetuning on homologous sequences is not a new idea; it’s been done since earlier ML for protein design works (Alley et al., 2020, Biswas et al., 2021) as well as recent works (Gordon et al., 2025). This limits the novelty of the work and the completeness of the discussion.
>
> Our novelty does not lie in fine-tuning and MCTS separately. AlphaDE is a novel framework that deeply integrates two state-of-the-art concepts: an AlphaZero-style search algorithm (MCTS) and a context-aware, fine-tuned PLM prior. We follow a principle of effective searching (i.e., AlphaZero-like MCTS) with strong prior guidance (i.e., fine-tuned protein language model), inspired by the natural large language model technique paradigm such as fine-tuning and test-time inference, to boost in-silicon directed evolution with a novel technique paradigm. Therefore, the integration of AlphaZero-style MCTS with fine-tuned PLM prior is a conceptual and algorithmic leap beyond prior approaches.
>
> 3.	Computational costs is a lot higher. Appendix L reports that AlphaDE takes 4.74 hours, vs 0.69 hours for EvoPlay.
>
> As AlphaDE utilizes the protein language model, it is expected to take a longer time to run. We believe this cost-to-performance trade-off is favorable and justified for the goal of identifying high-fitness protein variants. The runtime remains practical for in-silicon design cycles, which are orders of magnitude faster and cheaper than wet-lab experiments.
>
> 4.	The simulated landscapes are very toy settings, limiting its applicability to the real world. This is inherent to fitness landscape exploration type works, and not unique to this paper, but nonetheless limits the ultimate impact.
>
> Our study follows the established and widely adopted paradigm in the directed evolution community, where rigorous in-silicon benchmarking with fitness oracles is a critical step for evaluating novel algorithms. We fully agree that experimental validation is the ultimate measure of success. However, as wet-lab experiments are exceptionally resource-intensive and time-consuming, the primary goal of this work is to introduce and rigorously validate a powerful new algorithmic framework. We have explicitly added the need for future experimental validation in the Limitation section of our revised manuscript.
>
> 5.	IIUC from the Appendix C2, the process stops when the current mutated sequence fitness is lower than the wildtype. Is that overly greedy? If you take out this termination requirement, how often would you see the fitness climb back up? Given how rugged protein landscapes are, this is type of investigation might be interesting.
>
> Yes, this termination condition is greedy, which is reasonable as we want to maximize the fitness. At the same time, there is another termination condition that when the state sequence remains unchanged or changes to a previously generated sequence, the current episode terminates. This enforces AlphaDE to explore new sequences and helps avoid repetitive local search. Meanwhile, we do observe that the fitness climbs back up very often, which is expected as protein landscapes are rugged.

---

### Official Review · Reviewer_j9JA · 2025-11-06

**Soundness:** 3
**Presentation:** 3
**Contribution:** 2
**Rating:** 4
**Confidence:** 5

**Summary:**

This paper proposes AlphaDE, an in-silico directed evolution framework that integrates a fine-tuned protein language model (PLM) with Monte Carlo Tree Search (MCTS) for protein sequence optimization. The PLM is fine-tuned on homologous protein sequences to learn domain-specific evolutionary constraints, while the MCTS performs iterative exploration of the sequence space to identify beneficial mutations guided by the model’s probabilities. Experiments on eight benchmark protein engineering tasks demonstrate that AlphaDE achieves higher fitness improvements than several baselines, including TreeNeuralTS, TreeNeuralUCB, PEX, and AdaLead, even under limited fine-tuning data.

**Strengths:**

- The paper is well written and clearly structured, providing both background and algorithmic details.

- Combining fine-tuned PLMs with MCTS is conceptually sound and leverages recent progress in both protein modeling and search algorithms.

- Strong empirical evaluation on multiple protein datasets with reproducible settings.

- Demonstrates few-shot fine-tuning results, suggesting potential data efficiency.

**Weaknesses:**

- Limited novelty: The central idea closely overlaps with existing works on ML-guided directed evolution using protein LMs, particularly "Protein Design by Directed Evolution Guided by Large Language Models" (IEEE Transactions on Evolutionary Computation) [1], which already proposed LLM-based mutation guidance; and "LatentDE: Latent-based Directed Evolution for Protein Sequence Design" (Machine Learning: Science and Technology) [2], which introduced latent-space optimization for protein design using pretrained models. The proposed fine-tuning and search mechanisms appear incremental rather than fundamentally new.

- The integration of tree search does not significantly advance beyond prior reinforcement-learning or latent-search frameworks (e.g., LatentDE).

- Lacks biological validation or wet-lab evidence to confirm improved protein functionality.

- No clear ablation to isolate contributions of PLM fine-tuning vs. MCTS itself.

- Evaluation largely depends on oracle models; real-world applicability remains uncertain.

*** References:

[1] Trong Thanh Tran and Truong-Son Hy, Protein Design by Directed Evolution Guided by Large Language Models, IEEE Transactions on Evolutionary Computation (Q1, Impact Factor = 14.3), vol. 29, no. 2, pp. 418-428, April 2025, DOI 10.1109/TEVC.2024.3439690.
URL: https://ieeexplore.ieee.org/document/10628050

[2] Thanh V. T. Tran, Nhat Khang Ngo, Viet Thanh Duy Nguyen, and Truong-Son Hy, LatentDE: Latent-based Directed Evolution for Protein Sequence Design, Machine Learning: Science and Technology (Q1, Impact Factor = 6.3), Volume 6, Number 1, DOI 10.1088/2632-2153/adc2e2.
URL: https://iopscience.iop.org/article/10.1088/2632-2153/adc2e2/pdf

**Questions:**

How does the proposed AlphaDE fundamentally differ in principle or expected outcome from prior LLM-guided directed evolution frameworks such as "Protein Design by Directed Evolution Guided by Large Language Models" [1] and "LatentDE" [2]? Specifically, could you clarify what new insights or capabilities are gained by combining fine-tuned PLMs with Monte Carlo Tree Search beyond improved sampling efficiency?

---

> ### Author Response · Authors · 2025-11-18
> **Rebuttal part I**
>
> We sincerely appreciate Reviewer j9JA for the detailed comments. We provide the following clarifications.
>
> 1.	Limited novelty: The central idea closely overlaps with existing works on ML-guided directed evolution using protein LMs, particularly "Protein Design by Directed Evolution Guided by Large Language Models" (IEEE Transactions on Evolutionary Computation) [1], which already proposed LLM-based mutation guidance; and "LatentDE: Latent-based Directed Evolution for Protein Sequence Design" (Machine Learning: Science and Technology) [2], which introduced latent-space optimization for protein design using pretrained models. The proposed fine-tuning and search mechanisms appear incremental rather than fundamentally new.
>
> Although MLDE [1] and LatentDE [2] utilize pretrained protein language models to pinpoint the mutation guidance, they use simple searching or optimization techniques. Specifically, MLDE [1] uses beam search in the discrete sequence space based on an importance metric calculated by a k-mer's relevancy and entropy in a sequence. LatentDE [2] uses gradient-based method to search in the latent space represented by a frozen, general-purpose pretrained protein language model (i.e., ESM-2).
>
> The major advancements of AlphaDE over MLDE and LatentDE lie in two points. First, we use the one of the most powerful optimization techniques (AlphaZero-style MCTS) to learn to search. Unlike beam search or gradient ascent, MCTS performs a look-ahead search, simulating and evaluating multi-step mutation paths to strategically explore the fitness landscape. Second, we utilize the homology fine-tuning to specialize the general PLM into a domain-specific prior. This aligns the model's understanding with the evolutionary constraints of the target protein family, providing a much stronger and more relevant guide for the MCTS process. Therefore, AlphaDE is not an incremental improvement but a novel framework that deeply integrates two state-of-the-art concepts: a AlphaZero-style search algorithm (MCTS) and a context-aware, fine-tuned PLM prior. This combination enables a more intelligent and efficient exploration of the sequence space.
>
> At the same time, we add the comparison with MLDE [1] and LatentDE [2] as suggested by you. The empirical results strongly support this methodological advancement. AlphaDE clearly outperforms MLDE [1] and LatentDE [2], which also perform well by utilizing the pretrained protein language models, in the benchmark experiments.
>
> | Method | avGFP | AAV | TEM | E4B | AMIE | LGK | PAB1 | UBE2I |
> |---|---|---|---|---|---|---|---|---|
> | AlphaDE | **3.86** | **7.95** | **1.22** | **7.75** | **0.24** | **0.04** | **1.47** | 2.97 |
> | LatentDE [2] | 3.79 | 4.72 | 1.20 | 3.88 | -3.34 | -1.13 | 0.58 | 1.29 |
> | MLDE [3] | 2.04 | -3.38 | 0.08 | 3.84 | -1.38 | -0.19 | 0.92 | 2.75 |
>
>
> 2.	The integration of tree search does not significantly advance beyond prior reinforcement-learning or latent-search frameworks (e.g., LatentDE).
>
> First, we compare with LatentDE (latent-search framework, see results in Q1) and EvoPlay (reinforcement-learning framework, see results in Table 1). Results show that AlphaDE achieves the state-of-the-art performance.
>
> More fundamentally, the integration of AlphaZero-style MCTS with fine-tuned PLM prior is a conceptual and algorithmic leap beyond these prior approaches. Please see Q1 for details. To summarize, here we follow a principle of effective searching (i.e., AlphaZero-like MCTS) with strong prior guidance (i.e., fine-tuned protein language model), inspired by the natural large language model technique paradigm such as fine-tuning and test-time inference, to boost in-silicon directed evolution with a novel technique paradigm. We do believe this is a significantly advance in terms of both the concept and algorithm parts for directed evolution.

---

> > ### Author Response · Authors · 2025-11-18
> > **Rebuttal part II**
> >
> > 3.	Lacks biological validation or wet-lab evidence to confirm improved protein functionality.
> >
> > We acknowledge the reviewer's valid point regarding the importance of wet-lab validation. Our study follows the established and widely adopted paradigm in the directed evolution community, where rigorous in-silicon benchmarking with fitness oracles is a critical step for evaluating novel algorithms. In the avGFP condensing experiments, we utilize AlphaFold 3 for biological validation.
> >
> > We fully agree that experimental validation is the ultimate measure of success. However, as wet-lab experiments are exceptionally resource-intensive and time-consuming, the primary goal of this work is to introduce and rigorously validate a powerful new algorithmic framework. We have explicitly addressed the need for future experimental validation in the Limitation section of our revised manuscript.
> >
> > It is our sincere hope that the demonstrated effectiveness of AlphaDE will encourage and enable experimental biologists to adapt this method for their specific protein engineering challenges, thereby translating these promising in-silicon results into tangible biological applications.
> >
> > 4.	No clear ablation to isolate contributions of PLM fine-tuning vs. MCTS itself.
> >
> > The ablation study on both PLM fine-tuning and MCTS is provided in Appendix E. Table 4 in Appendix E compares a model learning from scratch, the base PLM model, and the fine-tuned PLM model. The fine-tuned model outperforms the base pre-trained PLM model and the mode learning from scratch. Table 5 in Appendix E compares the MCTS, greedy search, and beam search. MCTS is more effective than simpler search strategies like beam search or greedy search.
> >
> > 5.	Evaluation largely depends on oracle models; real-world applicability remains uncertain.
> >
> > Appendix F provides another benchmark experiments with ESM-1b oracle. Clearly, AlphaDE again beats various competitive baselines in the benchmark setting of EMS-1b oracle. At the same time, we stress that we design the algorithm for the general directed evolution and we hope biologists could tailor our AlphaDE for their own familiar experiments to explore real-world applicability.
> >
> > 6.	How does the proposed AlphaDE fundamentally differ in principle or expected outcome from prior LLM-guided directed evolution frameworks such as "Protein Design by Directed Evolution Guided by Large Language Models" [1] and "LatentDE" [2]? Specifically, could you clarify what new insights or capabilities are gained by combining fine-tuned PLMs with Monte Carlo Tree Search beyond improved sampling efficiency?
> >
> > Here we follow a principle of effective searching (i.e., AlphaZero-like MCTS) with strong prior guidance (i.e., fine-tuned protein language model), inspired by the natural large language model technique paradigm such as fine-tuning and test-time inference, to boost in-silicon directed evolution with a novel technique paradigm. Specifically, by formulating the directed evolution as a Markov Decision Process (MDP), our method explicitly models the sequential decision-making of selecting mutations. The integration of AlphaZero-like MCTS is crucial here, as it performs a look-ahead search to simulate and evaluate multi-step mutation paths, effectively planning a trajectory through sequence space. Furthermore, the fine-tuned PLM acts as a task-specific prior, constraining the search to a plausible local neighborhood around the parent sequence, which mirrors the natural evolutionary process of exploring proximal variants. This combination of iterative search with a localized prior makes AlphaDE particularly suited for directed evolution.

---

### Author Response · Authors · 2025-11-19
**New version of our paper during rebuttal**

Dear AC and Reviewers,

We have updated a new version of our paper during the rebuttal based on your comments and suggestions. More baselines, more related works, and more motivation descriptions are added! Enjoy this version and feel free to touch us!

Best,
The authors

---

### Note · Authors · 2026-01-07

I have read and agree with the venue's withdrawal policy on behalf of myself and my co-authors.